

# Non-invertible symmetries and higher representation theory II

Thomas Bartsch, Mathew Bullimore, Andrea E. V. Ferrari and Jamie Pearson

Department of Mathematical Sciences, Durham University,
Upper Mountjoy Campus, Stockton Road, Durham, DH1 3LE, United Kingdom

## Abstract

In this paper we continue our investigation of the global categorical symmetries that arise when gauging finite higher groups and their higher subgroups with discrete torsion. The motivation is to provide a common perspective on the construction of non-invertible global symmetries in higher dimensions and a precise description of the associated symmetry categories. We propose that the symmetry categories obtained by gauging higher subgroups may be defined as higher group-theoretical fusion categories, which are built from the projective higher representations of higher groups. As concrete applications we provide a unified description of the symmetry categories of gauge theories in three and four dimensions based on the Lie algebra $\mathfrak{so}(N)$, and a fully categorical description of non-invertible symmetries obtained by gauging a 1-form symmetry with a mixed 't Hooft anomaly. We also discuss the effect of discrete torsion on symmetry categories, based a series of obstructions determined by spectral sequence arguments.


# 1 Introduction

## 1.1 Background and motivation

Non-invertible topological defects in quantum field theory have long been known to exist in dimension $D = 2$ and are captured mathematically by fusion categories [1–23]. An important class of examples is obtained by gauging an anomaly-free finite symmetry group $G$, which leads to topological Wilson lines described by the fusion category $\mathsf{Rep}(G)$ of representations of $G$.

A generalisation of this construction is to gauge an anomaly-free subgroup of a finite group $G$ together with discrete torsion. This results in a rich class of symmetry categories known as a group-theoretical fusion categories, whose structure is entirely determined by the (projective) representation theory of finite groups [24–26]. These symmetry categories are in 1-1 correspondence with gapped boundary conditions of a three-dimensional Dijkgraaf Witten theory determined by the finite group $G$ and its anomaly.

The aim of this paper is to extend the above considerations to $D > 2$, building on our previous work [27] and the closely related work [28]. This is motivated by the remarkable recent progress on the existence and implications of non-invertible symmetries in higher dimensions [27–69]. A common thread of many constructions is to generate non-invertible symmetries by performing finite gauging procedures. Our idea is to provide a common framework for such examples that exhibits their full categorical structure in terms of higher representation theory.

On general grounds, finite symmetries in $D$ dimensions are expected to be captured mathematically by $(D-1)$-fusion categories. The latter encode the spectrum of topological operators of all dimensions $p = 1, \cdots, D-1$ as well as their fusion and braiding properties. For example, gauging a finite group symmetry $G$ is expected to result in a symmetry category $(D-1)\mathsf{Rep}(G)$ of $(D-1)$-representations of $G$. This symmetry category not only captures topological Wilson lines but also higher dimensional condensation defects that arise when gauging $G$.

In this paper, we explore the extension of this picture to the gauging of anomaly-free subgroups of higher groups in $D > 2$. This leads us to introduce the notion of group-theoretical higher fusion categories, which generate a rich class of non-invertible symmetries whose structure is determined by higher (projective) representation theory of finite higher groups. Many examples of non-invertible symmetries in $D > 2$ constructed thus far fall into this framework and may be understood in terms of higher analogues of standard results in the representation theory of higher groups.

Our approach will not be completely systematic and we will focus on dimensions $D = 2, 3, 4$. In $D = 4$, where the appropriate higher representation theory is less well-developed, input from known examples in the physics literature will provide an important guide. Some important applications are summarised below:

- A unified description of the symmetry categories of gauge theories in dimension $D = 3$ based on the Lie algebra $\mathfrak{so}(N)$ [70, 71], including disconnected global forms and discrete theta angles. They can be understood in terms of gapped boundary conditions for 4-dimensional Dijkgraaf-Witten theory with dihedral group $D_8$ symmetry. Similar considerations apply to gauge theories in $D = 4$.

- A categorical description of non-invertible symmetries obtained by gauging a 1-form symmetry with a mixed 't Hooft anomaly in dimension $D = 4$ [33]. These non-invertible symmetries are realised in $\mathcal{N} = 1$ supersymmetric Yang-Mills theories.[1] We explain how the dressing of symmetry defects with compensating anomalous TQFTs is a higher analogue of the appearance of projective representations in the representation theory of group extensions.

We also discuss of the effect of discrete torsion on the symmetry category, based a series of obstructions determined by spectral sequence arguments [72–75].

The approach will primarily build upon our previous work [27], which utilises the fact that finite symmetries can be gauged by summing over networks of symmetry defects. This may be used to define topological defects after gauging as topological defects before gauging together with instructions for how symmetry defects may end on them consistently. We will also discuss the connection to the approach in [28], where topological defects are defined by coupling to TQFT with the appropriate symmetry.

We will already encounter new phenomena in dimensions $D = 2, 3$ compared to our previous work due the appearance of projective higher representations. Correspondingly, we will emphasise the need to couple to TQFTs with anomalous symmetries in order to define topological defects.

In dimension $D = 4$, there are yet further new phenomena due to the fact that topological lines on a three-dimensional defect may braid. This is reflected in the richness of three-dimensional TQFTs or the existence of topological order described by SET phases in three dimensions. The mathematical structure of fusion 3-categories and 3-representation theory is less well-developed and we do not provide a completely systematic presentation in this case. In particular, we only consider 3-dimensional TQFTs of Turaev-Viro type, leaving a systematic treatment of more general TQFTs to future work. As a result, our formalism does not capture all possible topological defects in four dimensions that have been considered in the literature. We explain the appearance of TQFT-valued fusion coefficients in four dimensions and how they appear naturally when gauging 1-form symmetries with mixed anomalies.

## 1.2 Summary of results

The general setup this paper aims at is a quantum field theory $\mathcal{T}$ in $D$ dimensions with a finite group-like symmetry $G$. This could be at most a finite $(D-1)$-group and may have an 't Hooft anomaly specified by a cocycle $\alpha \in Z^{D+1}(G, U(1))$. The symmetry category is denoted $(D-1)\mathrm{Vec}^\alpha(G)$.

We then wish to gauge an anomaly free $(D-1)$-subgroup $H \subset G$. This requires choosing a trivialisation of the anomaly $\psi \in C^D(H, U(1))$ such that $\alpha|_H = (d\psi)^{-1}$, which may

---

[1] As explained below, our formalism only captures a subset of the topological defects constructed in [33].

be interpreted as a generalisation of discrete torsion. The resulting theory $\mathcal{T}/_\psi H$ has fusion $(D-1)$-category symmetry that we denote by

$$C(G, \alpha \,|\, H, \psi). \tag{1}$$

We refer to this as a higher group theoretical fusion category, which reduces to the standard notion of a group theoretical fusion category in dimension $D = 2$ [24–26]. For fixed $G, \alpha$, these symmetry categories are expected to arise on gapped boundary conditions in $(D + 1)$-dimensional Dijkgraaf-Witten theory based on the data $G, \alpha$, which then serves as a common symmetry TFT for this collection of symmetries.

We expect this construction to encompass a wide spectrum of interesting non-invertible symmetries in $D > 2$. In particular, in the case where $G$ is an ordinary group with 't Hooft anomaly $\alpha$, we show that the symmetries obtained by gauging a subgroup $H \subset G$ with discrete torsion $\psi$ correspond to so-called higher group-theoretical fusion categories $C(G, \alpha \,|\, H, \psi)$, whose structure is completely determined by the higher projective representation theory of $G$ and its subgroups:

$$C(G, \alpha \,|\, H, \psi) \;=\; \bigoplus_{[g] \in H \backslash G / H} (D-1)\mathsf{Rep}^{c_g(\alpha, \psi)}(H \cap {}^g H). \tag{2}$$

Here, the $D$-cocycle $c_g(\alpha, \psi)$ is constructed out of the 't Hooft anomlay $\alpha$ and the choice of discrete torsion $\psi$ and ${}^g H := gHg^{-1}$. The paper will explore aspects of this classification in dimensions $D = 2, 3, 4$.

We emphasise that underpinning these constructions is the $(D-1)$-fusion category

$$(D-1)\mathsf{Vec}, \tag{3}$$

which captures framed fully extended TQFTs in $(D-1)$ dimensions. For example, $(D-1)$-representations,

$$C(G \,|\, G) = (D-1)\mathsf{Rep}(G), \tag{4}$$

correspond to higher functors of the form $G \to (D-1)\mathsf{Vec}(G)$. The structure of $(D-1)\mathsf{Vec}$ and higher fusion categories more generally is not as well-developed for $D > 3$. We therefore restrict our attention to $D = 2, 3, 4$, and for $D = 4$ especially we will lean heavily upon physical considerations to light the way.

Let us summarise the content of each section as follows:

- In section 2 we review the gauging of anomaly-free subgroups of finite groups in two dimensions. The resulting symmetry categories are given by group-theoretical fusion categories, which are completely determined by the projective representation theory of ordinary groups. This is illustrated in two case studies.

- In section 3 we leverage the results from two dimensions to describe the gauging of anomaly-free 2-subgroups of finite 2-groups in three dimensions. The resulting symmetry categories are given by 2-group-theoretical fusion 2-categories, which are completely determined by the projective 2-representation theory of 2-groups. This is illustrated in two case studies.

- In section 4 we use the results from two and three dimensions to comment on aspects of gauging anomaly-free 3-subgroups of finite 3-groups in four dimensions. Our description will not be systematic, but will focus on highlighting important features in two case studies.

*Note added: During the course of this project, we were informed of potentially overlapping results by Lakshya Bhardwaj, Lea Bottini, Sakura Schäfer-Nameki and Apoorv Tiwari. We are grateful to them for coordinating the submission of our papers.*

## 2 Two dimensions

In this section, we review the gauging of subgroups of finite groups in two dimensions. We describe the associated fusion categories that capture the properties of topological lines after gauging. This will serve as a prototype whose structure we would like to emulate when gauging subgroups of higher groups in higher dimensions.

Underpinning the discussion of topological lines in two dimensions is the fusion category Vec, whose objects are finite-dimensional vector spaces $V = \mathbb{C}^n$ and whose morphisms are linear maps. Fusion is given by the tensor product of vector spaces. This may be considered a category of 1-dimensional TQFTs where morphisms correspond to topological interfaces and fusion corresponds to stacking.

Any topological line $L$ in two dimensions may be stacked with a decoupled 1-dimensional TQFT $V = \mathbb{C}^n$, which corresponds to taking the sum of $n$ identical copies of the line,

$$V \otimes L = L \oplus \cdots \oplus L = n \cdot L. \tag{5}$$

More formally, given any fusion category we may regard Vec as the sub-category that is generated by the identity line under fusion. As a consequence, the fusion rules of topological lines in two dimensions are understood to have integer coefficients.

### 2.1 Preliminaries

Let us consider a two-dimensional quantum field theory $\mathcal{T}$ with finite group symmetry $G$ and 't Hooft anomaly specified by a group cohomology class $[\alpha] \in H^3(G, U(1))$. Our convention is that a specification of $\mathcal{T}$ includes a choice of local counter term in background fields or equivalently a choice of representative $\alpha \in Z^3(G, U(1))$.

The symmetry category of $\mathcal{T}$ is the fusion category

$$\mathsf{Vec}^\alpha(G), \tag{6}$$

whose objects are finite-dimensional $G$-graded vector spaces and whose morphisms are grading preserving linear maps. Fusion is given by the tensor product of graded vector spaces with associator twisted by $\alpha \in Z^3(G, U(1))$. The symmetry category depends on the representative $\alpha$ only up to auto-equivalence.

The simple objects are vector spaces with a single graded component $V_g \cong \mathbb{C}$ and correspond to indecomposable topological lines generating the $G$ symmetry. They fuse according to the group law and satisfy an associativity relation as illustrated in figure 1.

### 2.2 Gauging groups

If the anomaly vanishes $[\alpha] = 0$, the symmetry $G$ may be gauged and the resulting theory $\mathcal{T}/G$ has symmetry category $\mathsf{Rep}(G)$. Its objects are finite-dimensional representations of $G$

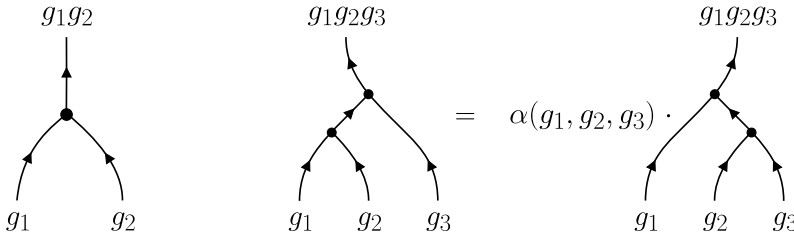

Figure 1

and its morphisms are intertwiners. Fusion is given by the tensor product of representations. The simple objects are given by irreducible representations of $G$ and are non-invertible if their dimension is greater than 1. There are a number of equivalent physical and mathematical interpretations of $\mathsf{Rep}(G)$:

- It captures topological Wilson lines in $\mathcal{T}/G$.

- It captures topological lines in $\mathcal{T}/G$ obtained by coupling to a 1-dimensional TQFT with $G$-action. This is the category of functors $G \to \mathsf{Vec}$, where $G$ is understood as a category with a single object, all of whose morphsims are invertible.

- It captures topological lines in $\mathcal{T}/G$ defined by topological lines in $\mathcal{T}$ together with instructions for how to intersect with networks of $G$ symmetry defects. This corresponds to defining lines in $\mathcal{T}/G$ as bi-modules for an algebra object in $\mathsf{Vec}(G)$.

The gauging procedure requires a choice of trivialisation $\alpha = (d\psi)^{-1}$ of the 't Hooft anomaly, where $\psi \in C^2(G, U(1))$ can be interpreted as discrete torsion. In order to keep track of this additional choice, we denote the resulting theory by $\mathcal{T}/_\psi G$. The effect of $\psi$ is to act by an auto-equivalence, so the symmetry category of $\mathcal{T}/_\psi G$ will be equivalent to $\mathsf{Rep}(G)$.

However, one may study topological interfaces between theories $\mathcal{T}/_{\psi_1} G$ and $\mathcal{T}/_{\psi_2} G$, which form projective representations of $G$ with 2-cocycle $\psi_1 - \psi_2$. The latter will also appear naturally when considering the gauging of an anomaly-free subgroup of an anomalous group as we will see in the following.

## 2.3 Gauging subgroups

The aim of this section is to generalise the above picture to gauging a general subgroup $H \subset G$. Let us then suppose the 't Hooft anomaly $\alpha$ is trivial on restriction to a subgroup $H$. This means there exists a 2-cochain $\psi \in C^2(H, U(1))$ such that

$$\alpha|_H = (d\psi)^{-1}. \tag{7}$$

This subgroup may then be gauged consistently by summing over appropriately weighted networks of topological line defects for $H \subset G$. The choice of trivialisation $\psi$ corresponds to gauging with a specified local counter term and is a generalisation of discrete torsion.

The result is a new theory $\mathcal{T}/_\psi H$ whose symmetry category we denote by[2]

$$\mathsf{C}(G, \alpha \,|\, H, \psi). \tag{8}$$

This is known as a group-theoretical fusion category [24–26]. The latter form an important class of fusion categories that generically have non-invertible simple objects and whose properties are determined by the projective representation theory of finite groups. In the remainder of this subsection, we summarise the construction of group-theoretical fusion categories from the perspective of topological line defects.

We note that the possible choices $H, \psi$ are in 1-1 correspondence with gapped boundary conditions for the 3-dimensional Dijkgraaf-Witten theory based on the data $G, \alpha$ [76, 77]. The latter then serves as the symmetry TFT for this collection of symmetries.

---

[2]When $\alpha$ or $\psi$ are trivial we often omit them in the symmetry category $\mathsf{C}(G, \alpha \,|\, H, \psi)$ of $\mathcal{T}/_\psi H$.

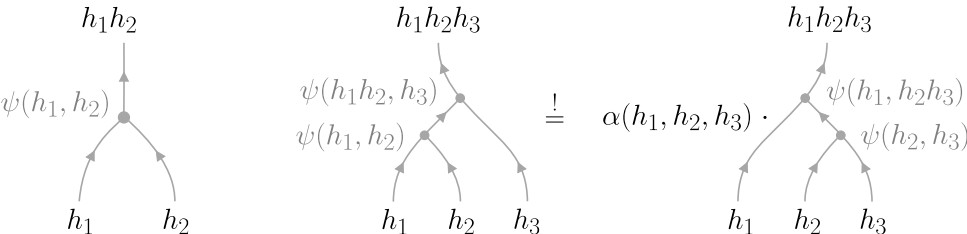

Figure 2

### 2.3.1 Objects

The starting point is the symmetry category $\mathsf{Vec}^\alpha(G)$ of $\mathcal{T}$. The 3-cocycle condition may be written explicitly as

$$(d\alpha)(g_1, g_2, g_3, g_4) \equiv \frac{\alpha(g_2, g_3, g_4)\,\alpha(g_1, g_2 g_3, g_4)\,\alpha(g_1, g_2, g_3)}{\alpha(g_1 g_2, g_3, g_4)\,\alpha(g_1, g_2, g_3 g_4)} \overset{!}{=} 1, \tag{9}$$

and ensures that the associator defines consistent relations when fusing four topological lines labelled by $g_1, g_2, g_3, g_4 \in G$ together. We will assume the 3-cocycle $\alpha$ is normalised in the sense that it is equal to 1 whenever one of its arguments is the identity element.

Let us now suppose that the anomaly becomes trivial upon restriction to $H \subset G$. This means that we can absorb the anomaly for $H$ by attaching phases $\psi(h_1, h_2) \in U(1)$ to junctions of topological lines labelled by $h_1, h_2 \in H$ as shown in figure 2. In order for these phases to cancel the anomaly, they need to satisfy

$$\frac{\psi(h_1 h_2, h_3)\,\psi(h_1, h_2)}{\psi(h_2, h_3)\,\psi(h_1, h_2 h_3)} \overset{!}{=} \alpha(h_1, h_2, h_3), \tag{10}$$

as shown on the right hand side of figure 2. This can be identified with the trivialisation condition $\alpha|_H \overset{!}{=} (d\psi)^{-1}$. Note that the choice of 2-cochain $\psi \in C^2(H, U(1))$ is not unique: shifting $\psi \to \psi \cdot \omega$ by any 2-cocycle $\omega \in Z^2(H, U(1))$ will lead to an equally valid trivialisation of $\alpha|_H$. We interpret this non-uniqueness as the freedom to add discrete torsion for the subgroup $H$.

Fixing a trivialisation $\psi$, we then gauge $H$ by summing over (equivalence classes of) networks of $H$-defects with phases $\psi(h_1, h_2)$ attached to junctions of topological lines. The result is a new theory $\mathcal{T}/_\psi H$ whose topological lines are constructed from topological lines in the ungauged theory $\mathcal{T}$ together with instructions for how networks of $H$-defects may intersect with them consistently. This is illustrated schematically in figure 3.

Concretely, we need to equip the topological defect with instructions for how networks of $H$-defects can end on it consistently from the left and from the right in a manner that is compatible with their topological nature. Mathematically, this reproduces the symmetry category $\mathsf{C}(G, \alpha \,|\, H, \phi)$ as the category of bimodules for an algebra object $A(H, \psi)$ in $\mathsf{Vec}^\alpha(G)$ associated to $H$ and $\psi$.

$$\left\langle \vphantom{\int} \right\rangle_{V \big/ \mathcal{T}/_\psi H} = \sum_{h_1, h_2, \dots} \left\langle \vphantom{\int} \right\rangle_{V \big/ \mathcal{T}}$$

Figure 3

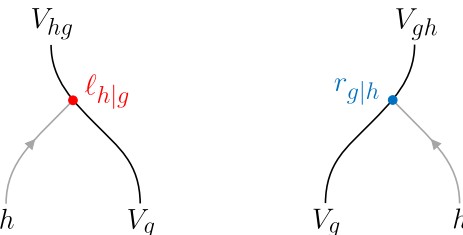

Figure 4

Let us start from a general topological line in $\mathcal{T}$ corresponding to an object of $\mathsf{Vec}^{\alpha}(G)$, which is a $G$-graded vector space $V = \oplus_g V_g$. Instructions for how symmetry defects $h \in H$ end on it from the left and right are specified by morphisms

$$\ell_{h|g} : h \otimes V_g \rightarrow V_{hg}, \qquad \text{and} \qquad r_{g|h} : V_g \otimes h \rightarrow V_{gh}, \tag{11}$$

as illustrated in figure 4.

The left and right morphisms must be compatible with fusion of symmetry defects in the bulk, which leads to the consistency conditions

$$\psi(h_1, h_2) \cdot \ell_{h_1 h_2 | g} = \alpha(h_1, h_2, g) \cdot \ell_{h_1 | h_2 g} \circ \ell_{h_2 | g}, \tag{12}$$

$$\psi(h_1, h_2) \cdot r_{g | h_1 h_2} = \alpha(g, h_1, h_2)^{-1} \cdot r_{g h_1 | h_2} \circ r_{g | h_1}, \tag{13}$$

illustrated in figures 5 and 6 respectively. In addition, the left and the right morphisms must be compatible with one another in the sense that

$$r_{h_1 g | h_2} \circ \ell_{h_1 | g} = \alpha(h_1, g, h_2) \cdot \ell_{h_1 | g h_2} \circ r_{g | h_2}, \tag{14}$$

which is illustrated in figure 7. Solutions of these equations define a bimodule for the algebra object $A(H, \psi)$ in $\mathsf{Vec}^{\alpha}(G)$ associated to $H$ and $\psi$.

In the remainder of this subsection, we collect some known information about simple objects, fusion and morphisms in the symmetry category $\mathsf{C}(G, \alpha \,|\, H, \psi)$.

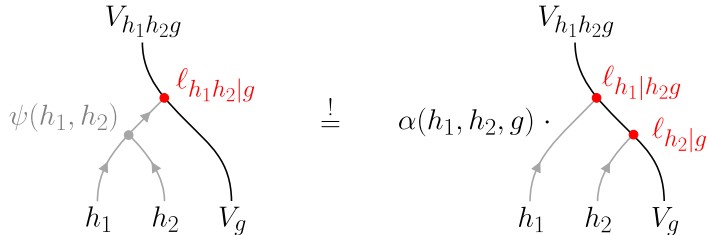

Figure 5

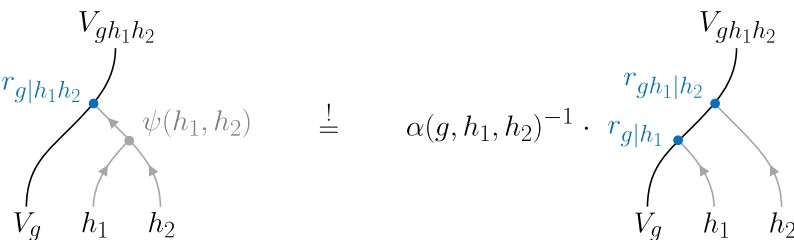

Figure 6

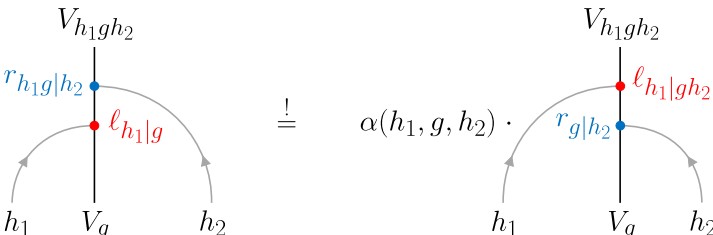

Figure 7

### 2.3.2 Simple Objects

From the form of the left and right morphisms in (11), it is clear that any solution will decompose as a direct sum of solutions supported on double $H$-cosets in $G$. Let us therefore restrict our attention to a solution supported on a single double coset $[g] \in H\backslash G/H$ with representative $g \in G$.

The associated vector space $V_g$ carries a projective representation $\Phi_g$ of the subgroup $H_g := H \cap {}^g H \subset H$ that is constructed from the left and right morphisms as

$$\Phi_g(h) := r_{hg|(h^g)^{-1}} \circ \ell_{h|g}, \tag{15}$$

where $h \in H_g$ and $h^g := g^{-1}hg$. From a physical point of view, group elements $h \in H_g$ correspond precisely to those elements in $H$ that leave $V_g$ invariant when intersecting it via $\Phi_g$ as illustrated in figure 8.

As a straightforward consequence of the consistency conditions (12), (13) and (14), the above defines a projective representation of $H_g$ in the sense that

$$\Phi_g(h_1 h_2) = c_g(h_1, h_2) \cdot \Phi_g(h_1) \circ \Phi_g(h_2), \tag{16}$$

for all elements $h_1, h_2 \in H_g$, where the 2-cocycle $c_g \in Z^2(H_g, U(1))$ depends on the anomaly $\alpha$ and its trivialisation $\psi$.

In order to bring the 2-cocycle of the projective representation into a more symmetric form, we redefine $\Phi_g \to \gamma_g \cdot \Phi_g$, where the 1-cochain $\gamma_g \in C^1(H_g, U(1))$ is given by the following combination

$$\gamma_g(h) = \psi(h^g, (h^g)^{-1})^{-1} \cdot \alpha(g, h^g, (h^g)^{-1})^{-1}. \tag{17}$$

It is straightforward to check using equation (13) that this redefinition is equivalent to using the alternative definition $\Phi_g := (r_{g|h^g})^{-1} \circ \ell_{h|g}$. This redefinition shifts the 2-cocycle $c_g \to c_g \cdot d\gamma_g$ and brings it into the form

$$c_g(h_1, h_2) := \frac{\psi(h_1^g, h_2^g)}{\psi(h_1, h_2)} \cdot \frac{\alpha(h_1, h_2, g)\,\alpha(g, h_1^g, h_2^g)}{\alpha(h_1, g, h_2^g)}. \tag{18}$$

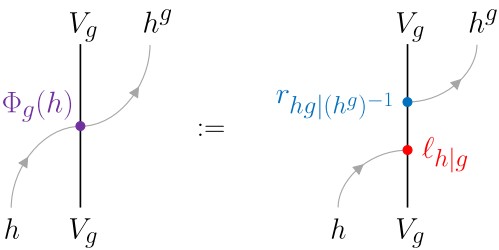

Figure 8

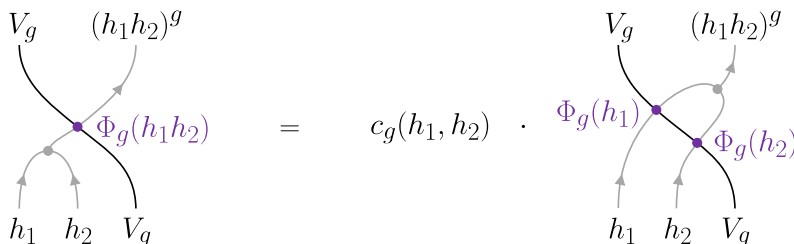

Figure 9

The interpretation of the projective representation is illustrated in figure 9, where it is shown to represent the compatibility with topological moves of the network of $H$-defects.

It is known that conversely such a projective representation determines a solution to the compatibility constraints for left and right morphisms [24, 26]. The above construction then sets up a bijection between isomorphism classes of simple objects and isomorphism classes of pairs $(g, \Phi_g)$ consisting of

1. A double coset $[g] \in H \backslash G / H$ with representative $g \in G$.

2. An irreducible projective representation $\Phi_g$ of $H_g$ with 2-cocycle

$$c_g(h_1, h_2) = \frac{\psi(h_1^g, h_2^g)}{\psi(h_1, h_2)} \cdot \frac{\alpha(h_1, h_2, g)\, \alpha(g, h_1^g, h_2^g)}{\alpha(h_1, g, h_2^g)} \,. \tag{19}$$

The isomorphism class of a simple object depends on the double coset representative $g$ and the 2-cocycle $c_g$ only up to isomorphism.

The above description of simple topological lines allows for the following alternative physical interpretation: Let us consider the line $g \in G$ in $\mathcal{T}$. This is left invariant under the action of $H_g \subset H$ and therefore supports a $H_g$ symmetry group. However, due to the bulk 't Hooft anomaly and its trivialisation, the topological line has an anomaly captured by the representative 2-cocycle $\bar{c}_g \in Z^2(H_g, U(1))$. In order to define a consistent topological line when gauging $H \subset G$, this anomaly must be cancelled by dressing with a 1-dimensional TQFT with $H_g$ symmetry and 't Hooft anomaly $c_g$. This is precisely specified by a vector space supporting a projective representation of $H_g$ with 2-cocycle $c_g$. It may simultaneously be regarded as a badly quantized Wilson line for $H_g$ whose anomalous transformation cancels that of the symmetry defect.

A similar mechanism will appear throughout and foreshadows many recent constructions of non-invertible symmetries in higher dimensions.

### 2.3.3 Morphisms

By similar reasoning, morphisms in the gauged theory $\mathcal{T}/_\psi H$ are obtained from morphisms in the original theory $\mathcal{T}$ together with compatibility conditions for how they intersect with networks of $H$-defects.

Concretely, given two simple objects $(g, \Phi_g)$ and $(g', \Phi_{g'})$, a morphism between them is obtained from a morphism $m : V_g \to V'_{g'}$ in $\mathsf{Vec}^\alpha(G)$ subject to compatibility conditions. First, since $m$ must preserve the grading of the vector spaces, such a morphism can only exist when $g = g'$. This is illustrated in figure 10.

In addition, the morphism $m$ must be compatible with topological manipulations of $H$-defects intersecting $V_g$ and $V_{g'}$ in the sense that

$$m \circ \Phi_g(h) \overset{!}{=} \Phi'_g(h) \circ m \,, \tag{20}$$

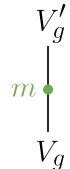

Figure 10

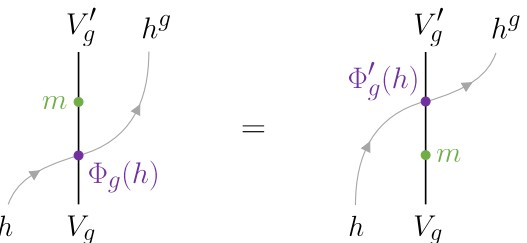

Figure 11

for all $h \in H_g$, which is illustrated in figure 11. We can thus identify morphisms in $\mathcal{T}/_\psi H$ with intertwiners between projective representations of $H_g$.

In summary, putting aside fusion, there is a decomposition,

$$\mathsf{C}(G, \alpha \,|\, H, \psi) \;\cong\; \bigoplus_{[g] \in H \backslash G / H} \mathsf{Rep}^{c_g}(H_g), \tag{21}$$

at the level of categories. A generic object of the symmetry category will thus be given by a collection of projective representations of subgroups $H_g \subset H$ with 2-cocycle $c_g$ indexed by (representatives of) double cosets $[g] \in H \backslash G / H$.

As a tautological example, consider the case where both $H$ and $\psi$ are trivial. Double cosets are then in 1-1 correspondence with group elements $g \in G$ and representations of the trivial group are finite-dimensional vector spaces. General objects can therefore be identified with $G$-graded vector spaces, reproducing the expected result,

$$\mathsf{C}(G, \alpha \,|\, 1) \;=\; \mathsf{Vec}^\alpha(G), \tag{22}$$

at the level of categories. In the other extreme, consider the case where $H = G$ with trivial anomaly. There is a single double coset with representative 1 so that

$$\mathsf{C}(G \,|\, G, \psi) \;=\; \mathsf{Rep}(G) \tag{23}$$

at the level of categories as anticipated from the discussion in subsection 2.2.

### 2.3.4  Fusion

The fusion of objects is completely determined by the tensor product of bimodules for the algebra object $A(H, \psi)$ in $\mathsf{Vec}^\alpha(G)$. The fusion rules of simple objects can be determined explicitly and are a special case of the fusion rules in equivariantisations of fusion categories presented in [78]. We will not present the general formula, but restrict ourselves to some salient features.

Consider two objects $L_1$ and $L_2$ supported on double cosets $[g_1]$ and $[g_2]$ respectively. Their fusion should be such that one can consistently insert additional $H$-defects in between them as illustrated in figure 12, and will thus be supported on the decomposition of $[g_1] \cdot [g_2]$ into double cosets.

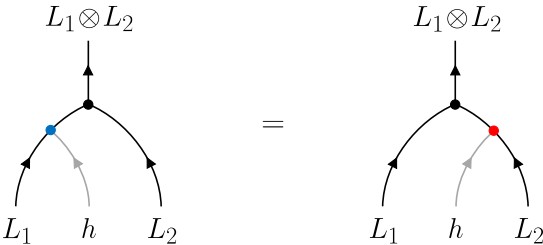

Figure 12

More generally, consider a generic object $L$ given by a collection $\{\Phi_g\}$ of projective representations indexed by representatives of double cosets $[g] \in H\backslash G/H$. We define the support of $L$ in the double coset ring $\mathbb{Z}[H\backslash G/H]$ by

$$\sup(L) := \sum_{[g]\in H\backslash G/H} \dim(\Phi_g) \cdot [g]. \tag{24}$$

Then the fusion of two objects $L$ and $L'$ must preserve their support in the sense that

$$\sup(L \otimes L') = \sup(L) * \sup(L'), \tag{25}$$

where $*$ denotes the ring structure on the double coset ring $\mathbb{Z}[H\backslash G/H]$. This can be defined explicitly as follows. First, given two double cosets $[g_1], [g_2]$, we can lift them to elements $x_1, x_2 \in \mathbb{Z}[G]$ by setting

$$x_i := \sum_{g \in [g_i]} 1 \cdot g \in \mathbb{Z}[G]. \tag{26}$$

Their product $x_1 \cdot x_2 \in \mathbb{Z}[G]$ is then $H$-invariant both from the left and from the right and hence determines a unique element in $\mathbb{Z}[H\backslash G/H]$ which we call $[g_1] * [g_2]$. The product of two generic elements in $\mathbb{Z}[H\backslash G/H]$ is obtained by linear extension.

In this way, the double coset ring forms the backbone of fusion with respect to the sum decomposition (21). The remaining fusion structure corresponds to decomposing and combining projective representations. We confine ourselves here to specific instances. A general formula can be found in [78].

## 2.4 Gauging extensions

Let us consider a group extension

$$1 \to A \to G \to K \to 1, \tag{27}$$

where $A$ is a finite abelian group and $K$ is a finite group. This is determined by a group homomorphism $\varphi : K \to \operatorname{Aut}(A)$ and an extension class $[e] \in H^2(K, A)$, where $A$ is understood as a $K$-module via the homomorphism. Any group element $g \in G$ may be expressed uniquely as a pair $(a, k) \in A \times K$ with multiplication is given by

$$(a_1, k_1) \cdot (a_2, k_2) := \left(a_1 \cdot {}^{k_1}a_2 \cdot e(k_1, k_2), k_1 \cdot k_2\right), \tag{28}$$

where we abbreviated ${}^k a := \varphi_k(a)$ for convenience.

If the short exact sequence splits (i.e. $[e] = 1$), this becomes a semi-direct product $G = A \rtimes_\varphi K$. Aspects of gauging the subgroups $A$ and $K$ in this case were summarised in the first instalment [27] and therefore we focus here on the orthogonal case of a non-trivial group extension with trivial action $\varphi$.

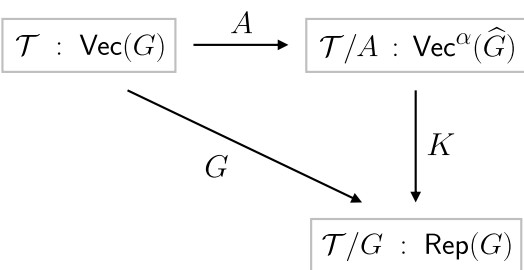

Figure 13

### 2.4.1 Gauging in steps

Let us thus consider a theory $\mathcal{T}$ with anomaly-free symmetry group $G$ of this kind. We gauge the symmetry $G$ in the absence of discrete torsion in two steps: we first gauge the subgroup $A$ and then subsequently gauge the remaining symmetry $K$. This is illustrated as a commutative diagram in figure 13.

- We start by gauging $A$ without discrete torsion, which corresponds to following the horizontal arrow in figure 13. We note that double cosets in $A\backslash G/A$ are in 1-1 correspondence with elements of $K$, so that simple objects after gauging are labelled by pairs $(\chi, k) \in \widehat{A} \times K$ with fusion

$$(\chi_1, k_1) \otimes (\chi_2, k_2) = (\chi_1 \cdot \chi_2, \, k_1 \cdot k_2). \tag{29}$$

The symmetry group of $\mathcal{T}/A$ can thus be identified with the product group $\widehat{G} = \widehat{A} \times K$. An explicit computation of the associator shows that this has a 't Hooft anomaly [9] given by the class $[\alpha] \in H^3(\widehat{G}, U(1))$ with cocycle representative

$$\alpha\big((\chi_1, k_1),(\chi_2, k_2),(\chi_3, k_3)\big) = \langle \chi_3, \, e(k_1, k_2) \rangle. \tag{30}$$

This may also be represented by the 3-dimensional SPT phase,

$$\int_X \widehat{\mathsf{a}} \cup \mathsf{k}^*(e), \tag{31}$$

in terms of the background fields $\widehat{\mathsf{a}} \in H^1(X, \widehat{A})$ and $\mathsf{k} : X \to BK$ for $\widehat{A}$ and $K$ respectively. In summary, the symmetry category of $\mathcal{T}/A$ is given by

$$\mathsf{C}(G|A) = \mathsf{Vec}^\alpha(\widehat{G}). \tag{32}$$

This is summarised in the top right of figure 13.

- We now gauge the remaining symmetry $K \subset \widehat{G}$ in $\mathcal{T}/A$, which corresponds to following the vertical arrow in figure 13. First, we note that double cosets of $K$ in $\widehat{G}$ are in 1-1 correspondence with elements $\chi \in \widehat{A}$, and that the corresponding 2-cocycle $c_\chi$ from (19) with $\alpha$ as in (30) and $\psi = 1$ reduces to

$$c_\chi(k_1, k_2) = \langle \chi, \, e(k_1, k_2) \rangle. \tag{33}$$

The simple objects are therefore labelled by pairs $(\chi, \Phi)$ consisting of

  1. a character $\chi \in \widehat{A}$,
  2. an irreducible projective representation $\Phi$ of $K$ with 2-cocycle $\langle \chi, e \rangle$.

Their fusion is determined by the multiplication of characters and the tensor product of projective representations,

$$(\chi_1, \Phi_1) \otimes (\chi_2, \Phi_2) = (\chi_1 \cdot \chi_2, \Phi_1 \otimes \Phi_2). \tag{34}$$

This has the following physical interpretation: Due to the mixed anomaly in $\mathcal{T}/A$ the topological line labelled by $\chi \in \widehat{A}$ has an anomaly under background gauge transformations for $K$ specified by $\langle \chi, e \rangle \in Z^2(K, U(1))$. To define a consistent topological line when gauging $K$, this must be absorbed by dressing with a 1-dimensional TQFT with the opposite anomaly, or equivalently a badly quantised Wilson line transforming in a projective representation of $K$.

Let us now check that following the above two steps sequentially is equivalent to following the diagonal arrow in figure 13, i.e. to gauging $G$ as a whole. The resulting symmetry category is known to be Rep$(G)$, which means that the simple objects after gauging $K$ should correspond to irreducible representations of $G$. Therefore let $(\chi, \Phi)$ be such a simple object, i.e. $\chi \in \widehat{A}$ is a character of $A$ and $\Phi : K \to \text{Aut}(V)$ is a projective representation of $K$ satisfying

$$\Phi(k_1 k_2) = \langle \chi, e(k_1, k_2) \rangle \cdot \Phi(k_1) \circ \Phi(k_2). \tag{35}$$

Using this data, we can define an action $\Psi$ of group elements $g = (a, k) \in G$ on $V$ by setting

$$\Psi(g)(v) := \overline{\chi(a)} \cdot \Phi(k)(v), \tag{36}$$

which can be checked to give a representation $\Psi : G \to \text{Aut}(V)$ of $G$ on $V$ satisfying

$$\Psi(g_1 \cdot g_2) = \Psi(g_1) \circ \Psi(g_2). \tag{37}$$

We claim this exhausts all irreducible representations of $G$. The fusion of simple objects corresponds to the tensor product of representations and morphisms are given by intertwiners. This reproduces the symmetry category

$$\mathsf{C}(G|G) = \mathsf{C}(\widehat{G}, \alpha | K) = \text{Rep}(G), \tag{38}$$

summarised in the bottom right of figure 13.

### 2.4.2 Adding discrete torsion

Let us now reconsider the previous example in the presence of discrete torsion.

First, consider gauging the entire symmetry $G$ with discrete torsion $\psi \in Z^2(G, U(1))$. We have already stated that this acts by an auto-equivalence of the symmetry category Rep$(G)$. This is compatible with the discussion above since the contributions from discrete torsion cancel out such that $c_e(g_1, g_2) = 1$ and simple objects are ordinary irreducible representations of $G$.

Now consider gauging $G$ in steps. We first gauge the abelian normal subgroup $A$ with discrete torsion $\phi \in Z^2(A, U(1))$. The simplest possibility is that this lifts to a discrete torsion for $G$. This means there exists a $\widetilde{\phi} \in Z^2(G, U(1))$ such that $[\phi] = \iota^*[\widetilde{\phi}]$, where $\iota : A \hookrightarrow G$ denotes the inclusion map in the short exact sequence. However, gauging $A$ with discrete torsion may produce an obstruction to subsequently gauging $K$ due to a symmetry extension or 't Hooft anomaly.

These obstructions are controlled by the Lyndon-Hochschild-Serre spectral sequence, which begins with

$$E_2^{p,q} = H^p(K, H^q(A, U(1))), \tag{39}$$

and converges to $H^{p+q}(G, U(1))$. This approach was discussed in [72–75] and is explored in more detail in the appendix A. The obstructions are organised in terms of the sequence of differentials $d_j^{0,2} : E_j^{0,2} \to E_j^{j,3-j}$ in the spectral sequence. The construction formalises the attempt to correct the topological terms in the action due to the relation $\delta a = k^*(e)$ satisfied by the background fields $a \in C^1(X, A)$ and $k : X \to BK$ for the $G$ symmetry. We consider the obstructions in turn:

- The first obstruction arises from the differential

$$d_2^{0,2} : H^2(A, U(1)) \to H^2(K, \widehat{A}). \tag{40}$$

This obstruction corresponds to a non-vanishing cohomology class

$$[f] := d_2^{0,2}([\phi]) \in H^2(K, \widehat{A}). \tag{41}$$

Note that due to the nilpotency $d_2^{2,1} \circ d_2^{0,2} = 0$ of the differential and its explicit form $d_2^{2,1} = [e] \cup (.)$, the obstruction must satisfy $[e] \cup [f] = 0 \in H^4(G, U(1))$. Upon choosing representatives $e$ and $f$, we are therefore always able to find a trivialisation $\omega \in C^3(K, U(1))$ such that $d\omega = e \cup f$.

This obstruction reflects the fact that the symmetry group $\widehat{G}$ of the gauged theory $\mathcal{T}/_\phi A$ will in general form a non-trivial extension

$$1 \to \widehat{A} \to \widehat{G} \to K \to 1, \tag{42}$$

with extension class $[f] \in H^2(K, \widehat{A})$ and 't Hooft anomaly $[\widehat{\alpha}] \in H^3(\widehat{G}, U(1))$ represented by the 3-dimensional SPT phase

$$\int_X \left[ \widehat{a} \cup k^*(e) - k^*(\omega) \right]. \tag{43}$$

Here, the inclusion of $\omega$ is needed to ensure that the SPT phases is still closed in light of the relation $\delta \widehat{a} = k^*(f)$ representing the fact that $\widehat{A}$ and $K$ form a non-trivial extension. The resulting symmetry category of $\mathcal{T}/_\phi A$ is therefore given by

$$C(G|A, \phi) = \mathsf{Vec}^{\widehat{\alpha}}(\widehat{G}). \tag{44}$$

Note that in the case of a vanishing first obstruction $[f] = 0$, the symmetry group $\widehat{G}$ reduces to a product group $\widehat{A} \times K$ as before. Furthermore, we can choose $\omega$ to be trivial in this case so that the corresponding anomaly $\widehat{\alpha}$ reduces to the anomaly $\alpha$ in (31).

- If the first obstruction vanishes (i.e. $[f] = 0$), there is a second obstruction coming from the differential

$$d_3^{0,3} : H^2(A, U(1)) \to H^3(K, U(1)). \tag{45}$$

This obstruction corresponds to a non-vanishing class

$$[\theta] := d_3^{0,3}([\phi]) \in H^3(K, U(1)). \tag{46}$$

In this case, gauging $A$ results in a theory $\mathcal{T}/_\phi A$ with symmetry group $\widehat{G} = \widehat{A} \times K$, whose anomaly is shifted by an additional pure anomaly $[\theta] \in H^3(K, U(1))$ that obstructs gauging $K$. The total anomaly is therefore represented by the 3-dimensional SPT phase

$$\int_X \left[ \widehat{a} \cup k^*(e) + k^*(\theta) \right], \tag{47}$$

and the corresponding symmetry category is given by

$$C(G|A, \phi) = \mathsf{Vec}^{\alpha+\theta}(\widehat{G}).$$ (48)

In the case of a vanishing second obstruction $[\theta] = 0$, there are no further obstructions so $K$ may be gauged. This is equivalent to gauging the entire symmetry group $G$ with discrete torsion given by the lift $\widetilde{\phi} \in H^2(G, U(1))$. The discrete torsion acts by an auto-equivalence of the symmetry category such that $C(G|G, \widetilde{\phi}) = \mathsf{Rep}(G)$.

## 2.5 Case study I

Let us consider $G = \mathbb{Z}_4$ viewed as an extension

$$1 \to \mathbb{Z}_2 \to \mathbb{Z}_4 \to \mathbb{Z}_2 \to 1,$$ (49)

with non-trivial class $[e] \in H^2(\mathbb{Z}_2, \mathbb{Z}_2)$. If we denote the generators of $A = \mathbb{Z}_2$ and $K = \mathbb{Z}_2$ by $x$ and $y$ respectively, the normalised 2-cocycle $e$ is completely determined by the condition $e(y, y) = x$.

We consider a theory $\mathcal{T}$ with symmetry group $G = \mathbb{Z}_4$ and trivial 't Hooft anomaly. There is no possibility for discrete torsion since $H^2(\mathbb{Z}_4, U(1)) = 0$. Gauging the whole symmetry $G$ leads to a theory $\mathcal{T}/G$ with symmetry category

$$C(\mathbb{Z}_4|\mathbb{Z}_4) = \mathsf{Rep}(\mathbb{Z}_4) \cong \mathsf{Vec}(\mathbb{Z}_4).$$ (50)

Alternatively, we may gauge the symmetry in steps by first gauging $A = \mathbb{Z}_2$ and subsequently gauging $K = \mathbb{Z}_2$. This example serves as a prototype for more interesting constructions in higher dimensions.

- First gauging $A = \mathbb{Z}_2$ results in a theory $\mathcal{T}/A$ with symmetry group $\widehat{G} = \widehat{A} \times K = \mathbb{Z}_2 \times \mathbb{Z}_2$ and mixed anomaly $\alpha \in Z^3(\mathbb{Z}_2 \times \mathbb{Z}_2, U(1))$ determined by the extension class $[e]$. This anomaly may be represented by the SPT phase,

$$\frac{1}{2}\int_X \widehat{a} \cup k \cup k,$$ (51)

in terms of the background fields $\widehat{a}, k \in H^1(X, \mathbb{Z}_2)$ for $\widehat{G}$. There is no possibility for discrete torsion since $H^2(\mathbb{Z}_2, U(1)) = 1$. The symmetry category of $\mathcal{T}/A$ is thus

$$C(\mathbb{Z}_4|\mathbb{Z}_2) = \mathsf{Vec}^\alpha(\mathbb{Z}_2 \times \mathbb{Z}_2).$$ (52)

- Now consider gauging $K = \mathbb{Z}_2$, which again does not allow for discrete torsion. The simple objects are labelled by pairs $(\chi, \Phi)$, where $\chi \in \widehat{A}$ and $\Phi$ is an irreducible projective representation of $K$ with 2-cocycle $\langle \chi, e \rangle$. Let us denote the generators of $\widehat{A} = \mathbb{Z}_2$ and $\widehat{K} = \mathbb{Z}_2$ by $\widehat{x}$ and $\widehat{y}$, respectively. For $\chi = 1$, we obtain two simple objects

$$U_0 := (1, 1), \quad \text{and} \quad U_2 := (1, \widehat{y}).$$ (53)

For $\chi = \widehat{x}$, we obtain two additional simple objects

$$U_3 := (\widehat{x}, f), \quad \text{and} \quad U_1 := (\widehat{x}, f \cdot \widehat{y}),$$ (54)

where the normalised 1-cochain $f : K \to U(1)$ is defined by $f(y) = i$. Using $f^2 = \widehat{y}$, the fusion of the simple objects can then be determined to be

$$(U_1)^n = U_{n \bmod 4}.$$ (55)

This reproduces the symmetry category $C(\mathbb{Z}_2 \times \mathbb{Z}_2, \alpha|\mathbb{Z}_2) = \mathsf{Vec}(\mathbb{Z}_4)$, which agrees with that of $\mathcal{T}/G$.

## 2.6 Case study II

Consider a theory $\mathcal{T}$ with anomaly free symmetry given by the dihedral group of order eight $G = D_8$. We systematically gauge subgroups $H \subset D_8$ with discrete torsion. The possible choices are in 1-1 correspondence with gapped boundary conditions for the 3-dimensional $D_8$ Dijkgraaf-Witten theory with trivial topological action, which plays the role of a symmetry TFT.

In two dimensions, an example is the orbifold branch of the $c = 1$ CFT or $\mathbb{Z}_2$-orbifold theory. In addition to the symmetry group $G = D_8$ considered here, this theory has a rich spectrum of non-invertible topological defects due to the fact that it is invariant under gauging of various subgroups [20]. We therefore emphasise that the symmetry categories discussed below form only part of the full fusion category symmetry in this example. Our considerations will also serve as a prototype for gauge theories with Lie algebra $\mathfrak{so}(N)$ in three and four dimensions, which will be considered in 3.6 and 4.6 respectively.

It is convenient to introduce generators $r, s$ of $D_8$ corresponding to rotation by $\pi/2$ and reflection such that

$$D_8 = \langle r, s \,|\, r^4 = s^2 = 1, \, srs^{-1} = r^{-1} \rangle, \tag{56}$$

which manifests its presentation as a semi-direct product $\mathbb{Z}_4 \rtimes \mathbb{Z}_2$. Alternatively, one may introduce generators $a := rs$ and $b := sr$ such that

$$D_8 = \langle a, b, s \,|\, a^2 = b^2 = s^2 = 1, \, ab = ba, \, sas^{-1} = b \rangle, \tag{57}$$

which manifests its presentation as a semi-direct product $D_4 \rtimes \mathbb{Z}_2$, where we denoted by $D_4 = \mathbb{Z}_2 \times \mathbb{Z}_2$ the dihedral group of order four.

The automorphism group of $D_8$ is again $D_8$: There is a $D_4$ subgroup of inner automorphisms generated by the conjugations $x \mapsto {}^{rs}x$ and $x \mapsto {}^{s}x$ as well as a $\mathbb{Z}_2$ subgroup of outer automorphisms generated by the automorphism that sends $r \mapsto r^3$ and $s \mapsto rs$. The latter acts on $D_4$ by sending ${}^{rs}(.) \mapsto {}^{s}(.)$, so that the total automorphism group is indeed given by $D_4 \rtimes \mathbb{Z}_2 \cong D_8$.

There are 10 subgroups $H \subset D_8$ forming 8 conjugacy classes, whose structure is summarised in figure 14. The subgroups are organized in rows according to their orders 1, 2, 4 and 8 from bottom to top. Normal subgroups are coloured in red whereas non-normal subgroups are coloured in black with red arrows indicating their transformation behaviour under conjugation. The encircled subgroup is the centre of $D_8$ and grey arrows denote inclusion as a normal subgroup. The blue arrow indicates the transformation behaviour of subgroups under the generator of outer automorphisms, which acts by reflection of the diagram.

The starting point is the symmetry category $C(D_8 | 1) = \mathsf{Vec}(D_8)$. We consider the symmetry categories that result from gauging subgroups with discrete torsion, beginning with subgroups of the smallest order and working upwards in figure 14.

### 2.6.1 Order two subgroups

We begin by gauging order 2 subgroups $H \cong \mathbb{Z}_2$. There is no possibility of discrete torsion since $H^2(\mathbb{Z}_2, U(1)) = 1$. There are 5 order 2 subgroups forming 3 conjugacy classes, two of which are related by an outer automorphism. Thus there are only two substantive cases to consider.

- The center $H = \langle r^2 \rangle \cong \mathbb{Z}_2$ of $D_8$ forms a non-split extension

$$1 \to \mathbb{Z}_2 \to D_8 \to D_4 \to 1, \tag{58}$$

with non-trivial extension class $[e] \in H^2(D_4, \mathbb{Z}_2)$. Gauging the center therefore leads to a symmetry group $\mathbb{Z}_2 \times D_4$ with 't Hooft anomaly determined by $[e]$, which can be

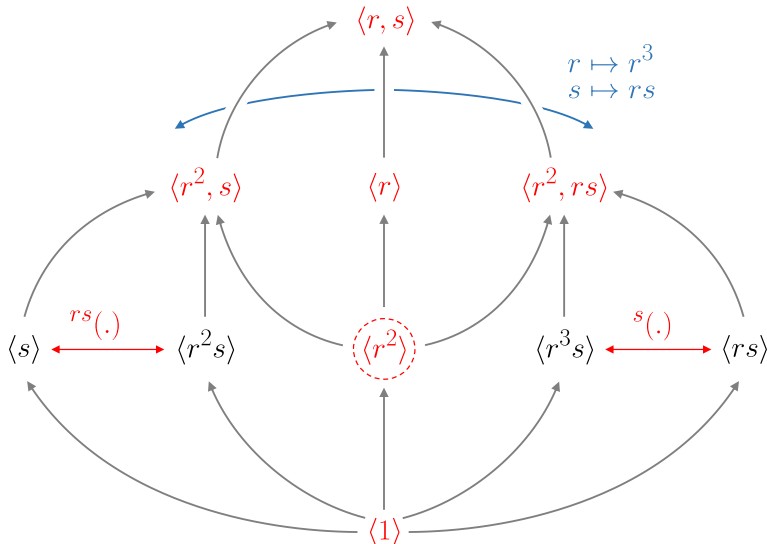

Figure 14

represented by the cubic SPT phase,

$$\frac{1}{2}\int_X \widehat{a}\cup a_1\cup a_2\,,\tag{59}$$

in terms of the background fields for $\mathbb{Z}_2\times D_4$. More concretely, we can describe the simple objects as follows: there are four double $H$-cosets $[1]$, $[r]$, $[s]$ and $[rs]$, all of whose stabilisers are given by $H$. The double coset ring is given by

$$[r]^2=[s]^2=[1]\,,\qquad [r]*[s]=[rs]\,.\tag{60}$$

There are therefore 8 simple objects corresponding to the following pairs of double cosets and irreducible representations

$$([1],\chi^n)\,,\qquad ([r],\chi^n)\,,\qquad ([s],\chi^n)\,,\qquad ([rs],\chi^n)\,,\tag{61}$$

where $n=0,1$ and $\chi$ denotes the generator of $\widehat{H}\cong\mathbb{Z}_2$. The fusion ring contains a $\mathbb{Z}_2$ subgroup generated by $C=([1],\chi)$ as well as a $D_4$ subgroup generated by $Y=([r],1)$ and $Z=([s],1)$, which commute with each other

$$C\otimes Y=Y\otimes C\,,\qquad C\otimes Z=Z\otimes C\,.\tag{62}$$

The symmetry can thus be identified with the product group $\mathbb{Z}_2\times D_4$ as stated above. The corresponding symmetry category is given by $\mathsf{C}(D_8\,|\,\langle r^2\rangle)=\mathsf{Vec}^\alpha(\mathbb{Z}_2\times D_4)$.

- Now consider the two non-normal subgroups $H=\langle s\rangle,\langle r^2 s\rangle\cong\mathbb{Z}_2$, which are related to each other by conjugation. For concreteness, consider gauging $H=\langle s\rangle$. There are three double cosets $[1]$, $[r]$, $[r^2]$ with stabilisers $H$, $1$, $H$ respectively. The double coset ring is given by

$$[r]*[r]=[1]+[r^2]\,,\qquad [r]*[r^2]=[r]\,,\qquad [r^2]*[r^2]=[1]\,.\tag{63}$$

There are therefore 5 simple objects corresponding to the following pairs of double cosets and irreducible representations

$$1=([1],1)\,,\quad U=([r^2],1)\,,\quad V=([1],\chi)\,,\quad W=([r^2],\chi)\,,\quad X=([r],1)\,,\tag{64}$$

where $\chi$ denotes the generator of $\widehat{H} \cong \mathbb{Z}_2$. The fusion ring contains a $D_4$ subgroup generated by $U$ and $V$ with $U \otimes V = W$ and additional relations

$$U \otimes X = X, \qquad V \otimes X = X, \qquad X \otimes X = 1 \oplus U \oplus V \oplus W. \tag{65}$$

The symmetry category is therefore a Tambara-Yamagami category of type $D_4$. A computation of the associator shows that $\mathsf{C}(D_8 \,|\, \langle s \rangle) = \mathsf{Rep}(D_8)$.

- Now consider the non-normal subgroups $H = \langle rs \rangle, \langle r^3 s \rangle \cong \mathbb{Z}_2$. They are related to each other by conjugation and to the subgroups in the previous bullet point by an outer-automorphism. The computation of the symmetry category is therefore the same up to relabelling, which implies $\mathsf{C}(D_8 \,|\, \langle rs \rangle) = \mathsf{C}(D_8 \,|\, \langle r^3 s \rangle) = \mathsf{Rep}(D_8)$.

### 2.6.2 Order four subgroups

There are three order 4 subgroups: one is isomorphic to $\mathbb{Z}_4$ and invariant under the outer automorphism, and the remaining two are isomorphic to $D_4$ and exchanged by the outer automorphism. In the latter case, there is the potential for discrete torsion because $H^2(D_4, U(1)) = \mathbb{Z}_2$. There are therefore only two substantive cases to consider.

- Consider gauging the normal subgroup $H = \langle r \rangle \cong \mathbb{Z}_4$. There are two double cosets, $[1]$ and $[s]$, both of which have $H$ as their stabiliser. The double coset ring is

$$[s] * [s] = [1]. \tag{66}$$

There are therefore 8 simple objects corresponding to the following pairs of double cosets and irreducible representations

$$([1], \chi^n), \qquad ([s], \chi^n), \tag{67}$$

where $n = 0, ..., 3$ and $\chi$ denotes the generator of $\widehat{H} \cong \mathbb{Z}_4$. The fusion ring is generated by $R := ([1], \omega)$ and $S := ([s], 1)$ subject to the relations

$$R^4 = S^2 = 1, \qquad S \otimes R \otimes S^{-1} = R^{-1}. \tag{68}$$

The symmetry can therefore be identified with the semi-direct product $\mathbb{Z}_4 \rtimes \mathbb{Z}_2 \cong D_8$, so that the corresponding symmetry category is given by $\mathsf{C}(D_8 \,|\, \langle r \rangle) = \mathsf{Vec}(D_8)$.

- Now consider the normal subgroup $H = \langle r^2, s \rangle \cong D_4$. There are again two double cosets $[1]$ and $[r]$, both of which have $H$ as their stabiliser. The double coset ring is

$$[r] * [r] = [1]. \tag{69}$$

There are therefore 8 simple objects corresponding to the following pairs of double cosets and irreducible representations

$$([1], \chi^n \omega^m), \qquad \text{and} \qquad ([r], \chi^n \omega^m), \tag{70}$$

where $n, m = 0, 1$ and $\chi, \omega$ denote the generators of $\widehat{H} \cong D_4$. The fusion ring is generated by $A := ([1], \chi)$, $B := ([1], \omega)$ and $D := ([r], 1)$ subject to the relations

$$A^2 = B^2 = D^2 = 1, \qquad D \otimes A \otimes D^{-1} = B. \tag{71}$$

The symmetry can therefore be identified with $D_4 \rtimes \mathbb{Z}_2 \cong D_8$ and the symmetry category is again given by $\mathsf{C}(D_8 \,|\, \langle r^2, rs \rangle) = \mathsf{Vec}(D_8)$.

Adding a discrete torsion element $\psi \in H^2(D_4, U(1)) = \mathbb{Z}_2$ leads to the same result, i.e. acts as an auto-equivalence of symmetry categories. This can be understood from the point of view of spectral sequences, interpreting $H^2(D_4, U(1))$ as $H^0(\mathbb{Z}_2, H^2(D_4, U(1)))$. Since there is no non-trivial group action of $\mathbb{Z}_2$ on $\mathbb{Z}_2$, $H^2(D_4, U(1))$ is a trivial $\mathbb{Z}_2$ module. We can then use the same arguments as in appendix A for split central extensions. It follows that there are no non-trivial differentials in the spectral sequence, which collapses at the second page. In particular, there is no obstruction in lifting $\psi$ to a class in $H^2(D_8, U(1))$.

- The normal subgroup $H = \langle r^2, rs \rangle \cong D_4$ is obtained from the bullet point above by an outer automorphism and therefore the computation of the symmetry category is the same up to relabelling. Adding discrete torsion again acts by an auto-equivalence of the symmetry category. We conclude that $C(D_8 \,|\, \langle r^2, s \rangle) = \text{Vec}(D_8)$.

Note that gauging both order four subgroups, including with discrete torsion, results in an identical symmetry category $\text{Vec}(D_8)$, up to equivalence. It is therefore possible that a theory $\mathcal{T}$ is invariant under gauging these subgroups, resulting in a rich spectrum of additional non-invertible duality defects that we have not not considered here. It was shown that this scenario is indeed realised when $\mathcal{T}$ is the $\mathbb{Z}_2$-orbifold CFT in [20].

### 2.6.3 Whole group

Finally, we gauge the entire symmetry group leading to the symmetry category $\text{Rep}(D_8)$. Adding a discrete torsion element $\psi \in H^2(D_8, U(1)) \cong \mathbb{Z}_2$ results in the same symmetry category up to equivalence. The results are summarised in figure 15.

There are various consistency checks on these results that correspond to taking different routes from bottom to top in figure 15. Due to the reflection symmetry of the diagram, it is sufficient to perform these checks for left hand side:

- Starting from the theory $\mathcal{T}$ with symmetry category $\text{Vec}(D_8)$ we can gauge the central subgroup $\langle r^2 \rangle \cong \mathbb{Z}_2$ to obtain the theory $\mathcal{T}/\langle r^2 \rangle$ whose symmetry category is given by

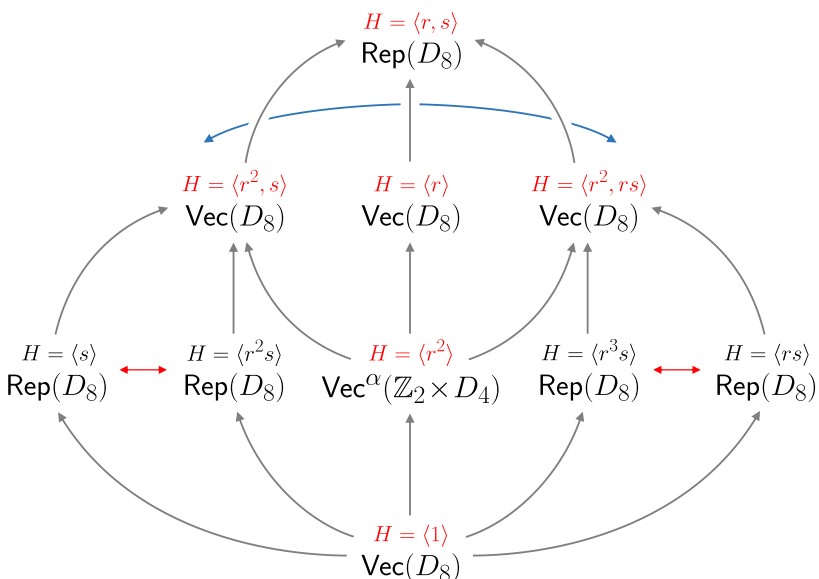

Figure 15

$\mathrm{Vec}^{\alpha}(\mathbb{Z}_2 \times D_4)$ as described in the first bullet point in 2.6.1. This contains a $D_4 \cong \mathbb{Z}_2 \times \mathbb{Z}_2$ subgroup generated by defects $Y, Z$, whose factors may be gauged independently:

- Gauging $\langle Y \rangle \cong \mathbb{Z}_2$ reproduces the theory $\mathcal{T}/\langle r \rangle$ with symmetry category given by $\mathrm{Vec}(D_8)$. The latter contains a $\mathbb{Z}_2$ subgroup generated by the defect $S$, whose gauging reproduces the theory $\mathcal{T}/\langle r, s \rangle$ with symmetry category $\mathrm{Rep}(D_8)$.

- Gauging $\langle Z \rangle \cong \mathbb{Z}_2$ reproduces the theory $\mathcal{T}/\langle r^2, s \rangle$ whose symmetry category is also $\mathrm{Vec}(D_8)$. The latter contains a $\mathbb{Z}_2$ subgroup generated by the defect $D$, whose gauging reproduces the theory $\mathcal{T}/\langle r, s \rangle$ with symmetry category $\mathrm{Rep}(D_8)$.

- Starting from $\mathcal{T}$ we can gauge the non-normal subgroup $\langle s \rangle \cong \mathbb{Z}_2$ to obtain the theory $\mathcal{T}/\langle s \rangle$ with symmetry category $\mathrm{Rep}(D_8)$ as described in the second bullet point in 2.6.1. The latter contains a $\mathbb{Z}_2$ subgroup generated by the defect $U$, whose gauging reproduces the theory $\mathcal{T}/\langle r^2, s \rangle$ with symmetry category $\mathrm{Vec}(D_8)$.

# 3 Three dimensions

In this section, we consider the gauging of 2-subgroups of finite 2-groups in three dimensions. We describe the associated symmetry categories that capture properties of topological defects after gauging. This will lead us to introduce the notion of a group-theoretical fusion 2-category, which is a natural generalisation of the structures that arose when gauging subgroups of groups in two dimensions in section 2.

Underpinning the description of topological surfaces in three dimensions is the fusion 2-category 2Vec, whose objects are finite-dimensional 2-vector spaces: Vec-module categories equivalent to $\mathrm{Vec}^n$ for some $n \geq 0$. This may be considered a category of 2-dimensional TQFTs where the integer $n \geq 0$ corresponds to the number of vacua and $\mathrm{Vec}^n$ is the category of boundary conditions.[3] A convenient representative is a 2-dimensional $\mathbb{Z}_n$ gauge theory, which we will denote by $\mathcal{Z}_n$ in the following. Fusion corresponds to stacking of 2-dimensional TQFTs.

Any topological surface defect $\mathcal{S}$ in three dimensions may be stacked with a decoupled 2-dimensional TQFT. From the discussion above, this corresponds to taking the sum of $n$ identical copies of the topological surface,

$$\mathcal{Z}_n \otimes \mathcal{S} = \mathcal{S} \oplus \cdots \oplus \mathcal{S} = n \cdot \mathcal{S}. \tag{72}$$

More formally, given any fusion 2-category we may regard 2Vec as the 2-subcategory generated by the identity topological surface under fusion. As a consequence, the fusion rules of topological surfaces in three dimensions may again be understood to have integer coefficients.

## 3.1 Preliminaries

Let us consider a three-dimensional quantum field theory $\mathcal{T}$ with finite group symmetry $G$ and 't Hooft anomaly specified by a group cohomology class $[\alpha] \in H^4(G, U(1))$. Our convention is again that a specification of $\mathcal{T}$ includes a choice of local counter term in background field or equivalently a choice of representative $\alpha \in Z^4(G, U(1))$.

---

[3]In addition, 2d TQFTs are also determined by a series of Euler terms that fix the theory's partition function on a 2-sphere. In the case of non-trivial Euler terms, the associated topoligical surface defect acts universally by non-trivial scalar multiplication on all local operators, including the identity operator. Since topological defects of this type are not interesting from a symmetry perspective, we only consider canonically normalised (stable) 2d TQFTs with trivial Euler terms (i.e. such that $\mathcal{Z}(S^2) = n$ for a TQFT specified by an integer $n \in \mathbb{Z}_{\geq 0}$) in what follows.

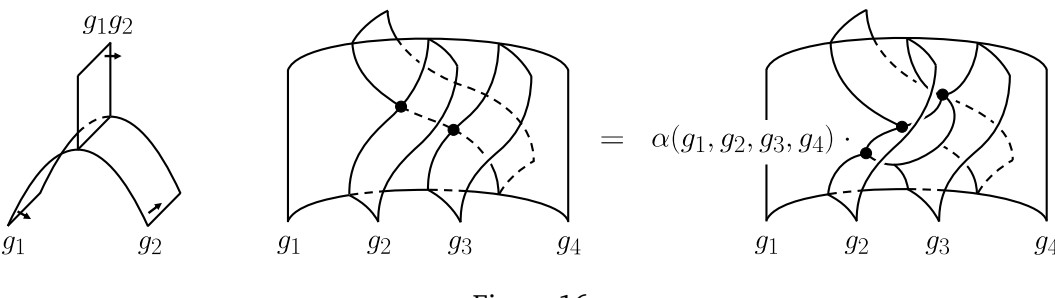

Figure 16

The symmetry category of $\mathcal{T}$ is the fusion 2-category

$$2\mathsf{Vec}^{\alpha}(G), \tag{73}$$

whose objects are finite-dimensional $G$-graded 2-vector spaces. The symmetry category depends on the representative $\alpha$ only up to auto-equivalence.

The simple objects are 2-vector spaces with a single graded component Vec attached to an element $g \in G$, and correspond to the indecomposable topological surfaces generating the $G$ symmetry. They fuse according to the group law and satisfy a pentagon relation as illustrated in figure 16.

## 3.2 Gauging groups

If the anomaly vanishes $[\alpha] = 0$, the symmetry $G$ may be gauged and the resulting theory $\mathcal{T}/G$ has symmetry category $2\mathsf{Rep}(G)$ [27,28]. There are a number of equivalent physical and mathematical interpretations of $2\mathsf{Rep}(G)$:

- It captures condensation defects for the topological Wilson lines in $\mathcal{T}/G$. This corresponds to the mathematical statement that $2\mathsf{Rep}(G) = \mathsf{Mod}(\mathsf{Rep}(G))$ is the idempotent completion of the delooping of $\mathsf{Rep}(G)$ [79].

- It captures topological surfaces in $\mathcal{T}/G$ obtained by coupling to a 2-dimensional TQFT with symmetry group $G$. Mathematically, $2\mathsf{Rep}(G)$ can be regarded as the 2-category of 2-pseudo-functors $G \to 2\mathsf{Vec}$, where $G$ is understood as a 2-category with a single object, all of whose morphisms are invertible.

- It captures topological surfaces in $\mathcal{T}/G$ defined by topological surfaces in $\mathcal{T}$ together with instructions for how to intersect with networks of $G$ symmetry defects. This corresponds to defining surfaces in $\mathcal{T}/G$ to be 2-bimodules for a certain 2-algebra object in $2\mathsf{Vec}(G)$.

For further mathematical background on 2-representations, we refer the reader to [80,81]. Independently of the interpretation, the simple objects are irreducible 2-representations, which can be labelled by the following concrete collection of data:

1. A subgroup $H \subset G$,

2. a 2-cocycle $c \in Z^2(H, U(1))$.

The equivalence class of the 2-representations only depends on the conjugacy class of the subgroup $H$ and the group cohomology class $[c] \in H^2(H, U(1))$. The physical interpretation is a topological surface on which the gauge symmetry is broken down to a subgroup and supplemented by a defect action corresponding to an SPT phase.

The gauging procedure requires a choice of trivialisation $\alpha = (d\psi)^{-1}$ of the 't Hooft anomaly, where $\psi \in C^3(G, U(1))$ can be interpreted as discrete torsion. We again denote the

resulting theory by $\mathcal{T}/_\psi G$. Up to equivalence, the symmetry category of $\mathcal{T}/_\psi G$ is independent of the choice of trivialisation $\psi$.

However, one may study topological interfaces between theories $\mathcal{T}/_{\psi_1} G$ and $\mathcal{T}/_{\psi_2} G$, which form projective 2-representations of $G$ with 3-cocycle $\psi_1 - \psi_2$. Similarly to before, irreducible projective 2-representations of this kind can be labelled by[4]

1. a subgroup $H \subset G$,

2. a 2-cochain $c \in C^2(H, U(1))$ satisfying $dc = (\psi_1 - \psi_2)|_H$.

They will also appear naturally when considering the gauging of an anomaly-free subgroup of an anomalous group as we will see in the following.

## 3.3 Gauging subgroups

The purpose of this section is to generalise the above picture to a general subgroup $H \subset G$. Let us then suppose the 't Hooft anomaly $\alpha$ is trivial upon restriction to a subgroup $H$. This means there exists a 3-cochain $\psi \in C^3(H, U(1))$ such that

$$\alpha|_H = (d\psi)^{-1}. \tag{74}$$

This subgroup may then be gauged consistently by summing over appropriately weighted networks of topological surface defects for $H \subset G$. The choice of trivialisation $\psi$ can again be recognised as a generalisation of discrete torsion.

In three dimensions, it is possible to generalise this construction further by gauging in the presence of a more general 3-dimensional TQFT corresponding to an SET phase with $H$ symmetry [82–84]. We will not consider this generalisation here, but return to a similar construction for 3-dimensional topological defects in four dimensions in section 4.

The result is a new theory $\mathcal{T}/_\psi H$ whose symmetry 2-category we denote by

$$C(G, \alpha | H, \psi). \tag{75}$$

We call this a group-theoretical fusion 2-category. We expect they form an interesting class of fusion 2-categories, which typically have non-invertible simple objects and whose properties are determined by the projective 2-representation theory of finite groups. In the remainder of this subsection, we summarise some elementary properties of group-theoretical fusion 2-categories from the perspective of topological surface defects.

We again note the possible choices are expected to correspond to gapped boundary conditions for the 4-dimensional Dijkgraaf-Witten theory based on $G, \alpha$ [85–87]. The latter then serves as the symmetry TFT for this collection of symmetries.

### 3.3.1 Objects

The starting point is the symmetry category $2\mathsf{Vec}^\alpha(G)$ of $\mathcal{T}$. The 4-cocycle condition may be written explicitly as

$$(d\alpha)(g_1, g_2, g_3, g_4, g_5) \equiv \frac{\alpha(g_2, g_3, g_4, g_5)\,\alpha(g_1, g_2 g_3, g_4, g_5)\,\alpha(g_1, g_2, g_3, g_4 g_5)}{\alpha(g_1 g_2, g_3, g_4, g_5)\,\alpha(g_1, g_2, g_3 g_4, g_5)\,\alpha(g_1, g_2, g_3, g_4)} \overset{!}{=} 1, \tag{76}$$

and ensures that the pentagonator defines consistent relations when fusing five topological surfaces labelled by $g_1, ..., g_5 \in G$ together. We again assume the 4-cocyle $\alpha$ is normalised.

---

[4]Note that for generic $\psi_1$ and $\psi_2$, a trivialisation $dc = (\psi_1 - \psi_2)|_H$ may not exist. In this case, there exists no projective 2-representation of $G$ labelled by the subgroup $H \subset G$.

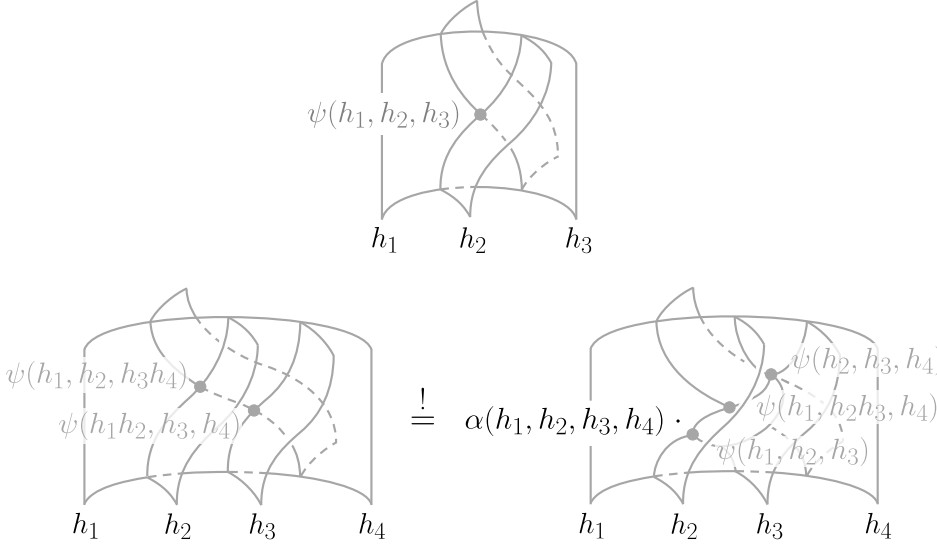

Figure 17

Let us now suppose that the anomaly is trivial upon restriction to $H \subset G$. This means that we can absorb the anomaly for $H$ by attaching phases $\psi(h_1, h_2, h_3) \in U(1)$ to junctions of topological surfaces labelled by $h_1$, $h_2$ and $h_3$ as shown in figure 17. In order for these phases to cancel the anomaly, they need to satisfy

$$\frac{\psi(h_1 h_2, h_3, h_4)\, \psi(h_1, h_2, h_3 h_4)}{\psi(h_2, h_3, h_4)\, \psi(h_1, h_2 h_3, h_4)\, \psi(h_1, h_2, h_3)} \stackrel{!}{=} \alpha(h_1, h_2, h_3, h_4), \tag{77}$$

as shown in the lower part of figure 17. This can be identified with the trivialisation condition $\alpha|_H \stackrel{!}{=} (d\psi)^{-1}$. Note that the choice of trivialisation $\psi$ is not unique, since adding 3-cocycles to $\psi$ will leave the trivialisation condition invariant. We again interpret this additional freedom as the possibility to add discrete torsion for the subgroup $H$.

Upon fixing a particular trivialisation $\psi$, we are then able to gauge $H$ by summing over (equivalence classes of) networks of $H$-defects with $\psi$ attached to junctions of topological surfaces. The result is a new theory $\mathcal{T}/_\psi H$, whose topological surfaces are constructed from topological surfaces in the ungauged theory $\mathcal{T}$ together with instructions for how networks of $H$-defects may intersect with them consistently. This is illustrated schematically in figure 18.

Concretely, we equip the topological surface $\mathcal{S}$ with instructions for how networks of $H$-defects can end on it consistently from the left and from the right in a manner that is compatible with their topological nature. This defines the objects of the symmetry category $C(G, \alpha \,|\, H, \phi)$ as 2-bimodules for a certain algebra object in the original symmetry category $2\mathsf{Vec}^\alpha(G)$ associated to $H$ and $\psi$.

Let us start from a general topological surface defect corresponding to an object of the

$$\left\langle \;\; \underset{\mathcal{S}}{\diagup} \;\; \right\rangle_{\mathcal{T}/_\psi H} = \sum_{h_1, h_2, h_3, \dots} \left\langle \;\; \underset{h_2 \quad\;\; h_3 \quad \mathcal{S}}{h_1 \quad \psi} \;\; \right\rangle_{\mathcal{T}}$$

Figure 18

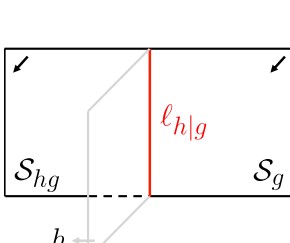
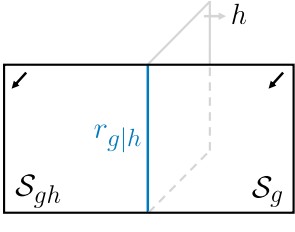

Figure 19

fusion 2-category $2\mathsf{Vec}^\alpha(G)$. This may be expressed as $\mathsf{Vec}^{\mathcal{S}}$ as a module category over Vec, where $\mathcal{S} = \bigoplus_g \mathcal{S}_g$ is a $G$-graded set. Concretely, writing $\mathcal{S}_g = \{1, \dots, n_g\}$ it corresponds to a general topological surface formed by sums of $n_g$ copies of the topological surface labelled by the group element $g \in G$.

Instructions for how symmetry defects $h \in H$ may end on it from left and right are specified by 1-morphisms

$$\ell_{h|g} : h \otimes \mathcal{S}_g \to \mathcal{S}_{hg}, \qquad \text{and} \qquad r_{g|h} : \mathcal{S}_g \otimes h \to \mathcal{S}_{gh}, \tag{78}$$

as illustrated in figure 19. In the following, we will call them left and right 1-morphisms respectively.

In addition, we need to give instructions for how the fusion of two symmetry defects $h, h' \in H$ in the bulk can end on $\mathcal{S}$ consistently from the left and from the right. This is implemented by 2-morphisms

$$\Psi^\ell_{h,h'|g} : \ell_{hh'|g} \Rightarrow \ell_{h|h'g} \otimes \ell_{h'|g}, \tag{79}$$

$$\Psi^r_{g|h,h'} : r_{g|hh'} \Rightarrow r_{gh|h'} \otimes r_{g|h}, \tag{80}$$

which we call the left and right 2-morphisms respectively. We also introduce left-right 2-morphisms

$$\Psi^{\ell r}_{h|g|h'} : r_{hg|h'} \otimes \ell_{h|g} \Rightarrow \ell_{h|gh'} \otimes r_{g|h'}, \tag{81}$$

describing how symmetry defects can end on $\mathcal{S}$ from the left and from the right at the same time. This is illustrated in figure 20.

The left and right 1- and 2-morphisms must be compatible with the fusion of symmetry defects in the bulk. This leads to the consistency conditions

$$\psi(h_1, h_2, h_3) \cdot \left[\Psi^\ell_{h_1, h_2|h_3 g} \otimes \ell_{h_3|g}\right] \circ \Psi^\ell_{h_1 h_2, h_3|g} \overset{!}{=} \alpha(h_1, h_2, h_3, g) \cdot \left[\ell_{h_1|h_2 h_3 g} \otimes \Psi^\ell_{h_2, h_3|g}\right] \circ \Psi^\ell_{h_1, h_2 h_3|g}, \tag{82}$$

and

$$\psi(h_1, h_2, h_3)^{-1} \cdot \left[\Psi^r_{gh_1|h_2, h_3} \otimes r_{g|h_1}\right] \circ \Psi^r_{g|h_1, h_2 h_3} \overset{!}{=} \alpha(g, h_1, h_2, h_3)^{-1} \cdot \left[r_{gh_1 h_2|h_3} \otimes \Psi^r_{g|h_1, h_2}\right] \circ \Psi^r_{g|h_1 h_2, h_3}, \tag{83}$$

which are illustrated in figure 21.

Similarly, the left-right 2-morphisms need to be compatible with the fusion of symmetry defects, which leads to the consistency conditions

$$\alpha(h_1, h_2, g, h_3)^{-1} \cdot \left[\Psi^\ell_{h_1, h_2|gh_3} \otimes r_{g|h_3}\right] \circ \Psi^{\ell r}_{h_1 h_2|g|h_3} \overset{!}{=} \left[\ell_{h_1|h_2 gh_3} \otimes \Psi^{\ell r}_{h_2|g|h_3}\right] \circ \Psi^{\ell r}_{h_1|h_2 g|h_3} \circ \left[r_{h_2 g|h_3} \otimes \Psi^\ell_{h_1, h_2|g}\right], \tag{84}$$

and

$$\alpha(h_1, g, h_2, h_3) \cdot \left[\ell_{h_1|gh_2 h_3} \otimes \Psi^r_{g|h_2, h_3}\right] \circ \Psi^{\ell r}_{h_1|g|h_2 h_3} \overset{!}{=} \left[\Psi^{\ell r}_{h_1|gh_2|h_3} \otimes r_{g|h_2}\right] \circ \Psi^{\ell r}_{h_1|g|h_2} \circ \left[\Psi^r_{h_1 g|h_2, h_3} \otimes \ell_{h_1|g}\right], \tag{85}$$

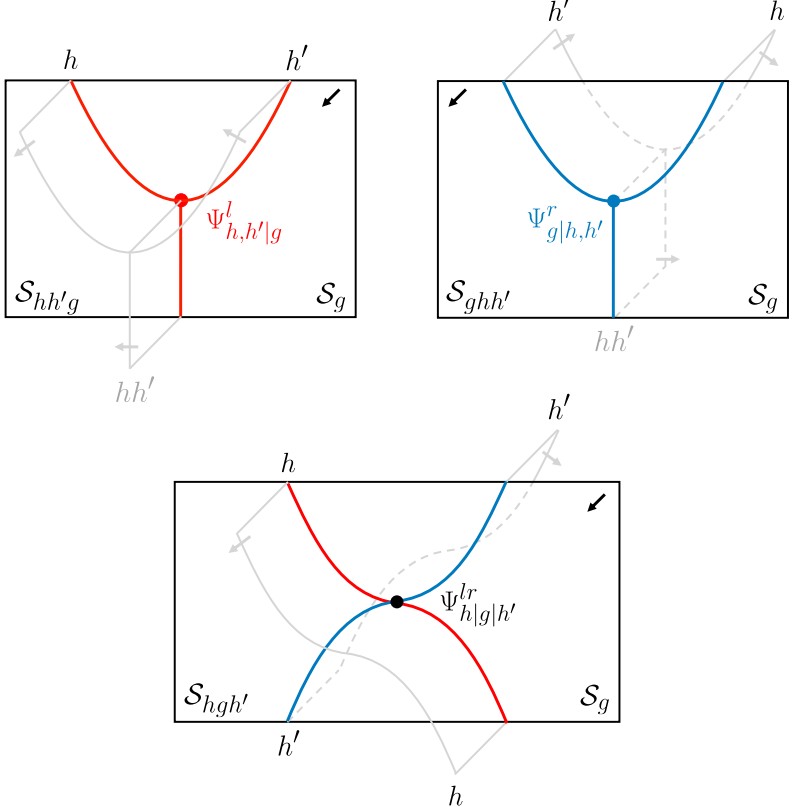

Figure 20

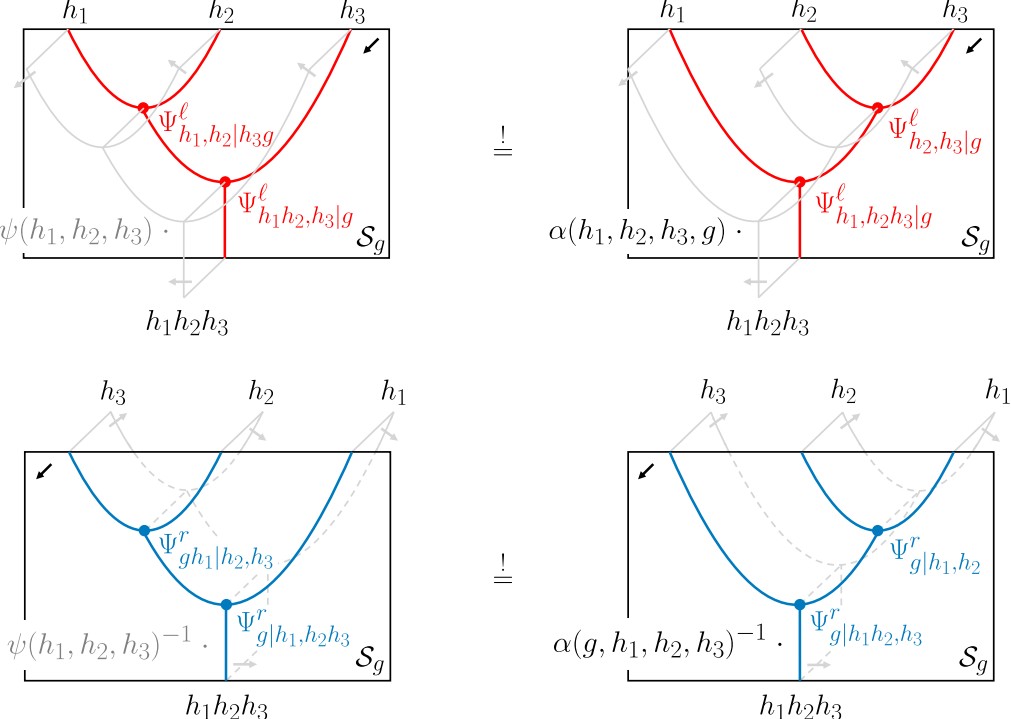

Figure 21

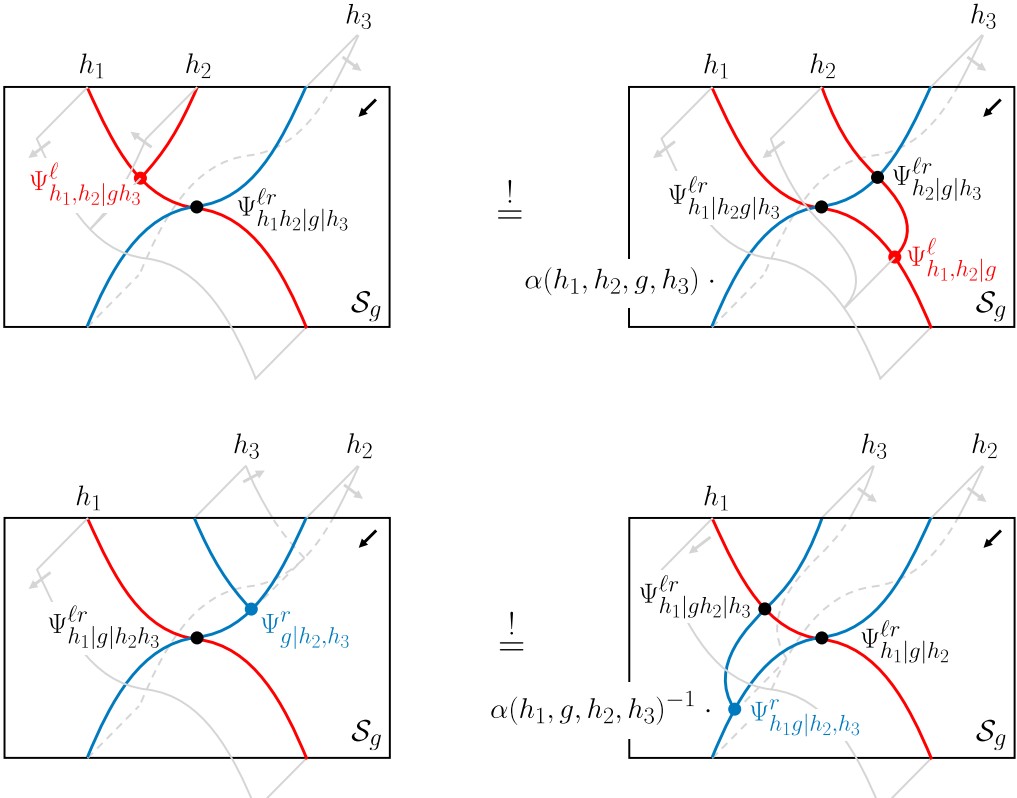

Figure 22

as illustrated in figure 22. Solutions to these equations define a 2-bimodule for the algebra object $A(H, \psi)$ in $2\mathrm{Vec}^\alpha(G)$ determined by $H$ and $\psi$.

In the remainder of this subsection, we derive some information about simple objects, fusion and morphisms in the symmetry 2-category $\mathsf{C}(G, \alpha \,|\, H, \psi)$.

### 3.3.2 Simple Objects

From the form of the left and right morphisms (78), it is clear that any solution will decompose as a direct sum of solutions supported on double $H$-cosets in $G$. Let us therefore restrict our attention to a solution supported on a single double coset $[g] \in H\backslash G/H$ with representative $g \in G$.

The associated set $\mathcal{S}_g$ carries a projective 2-representation $\Psi_g$ of the subgroup $H_g := H \cap {}^g H \subset H$ that is constructed from the left and right 1- and 2-morphisms as follows. First, we define 1-morphisms

$$\rho_g(h) \ := \ r_{hg|(h^g)^{-1}} \circ \ell_{h|g}, \tag{86}$$

with $h \in H_g$ and $h^g := g^{-1}hg$, which describe how symmetry defects pierce through $\mathcal{S}_g$ as illustrated in figure 23.

Next, we introduce 2-morphisms

$$\Psi_g(h, h') \ := \ \Psi^{\ell r}_{h|h'g|(h'^g)^{-1}} \circ \left[ \Psi^r_{hh'g|(h'^g)^{-1},(h^g)^{-1}} \otimes \Psi^\ell_{h,h'|g} \right], \tag{87}$$

that describe how the fusion of two symmetry defects in the bulk pierces through $\mathcal{S}_g$ as illustrated in figure 24.

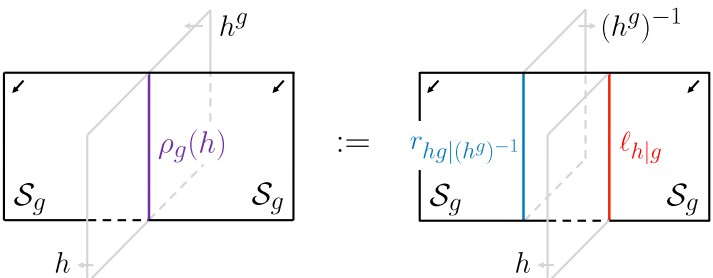

Figure 23

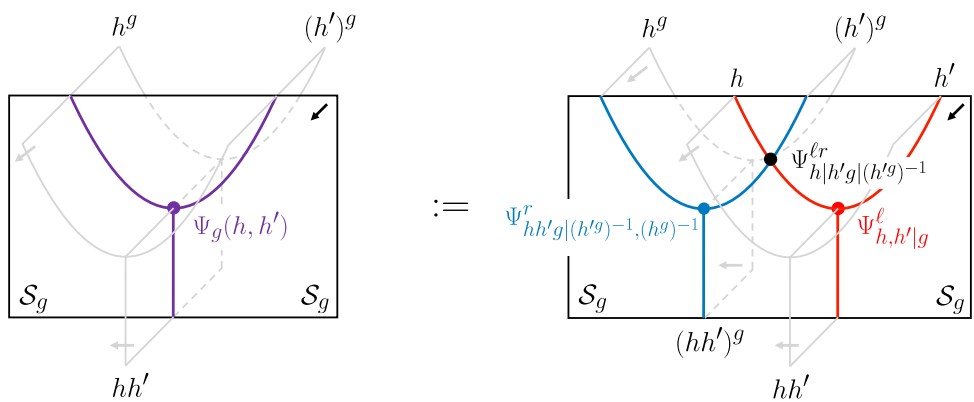

Figure 24

Using the consistency conditions (82), (83), (84) and (85), one can then check that the collection of 1-morphisms and 2-morphisms,

$$\Psi_g(h,h') : \ \rho_g(hh') \ \Rightarrow \ \rho_g(h) \otimes \rho_g(h'), \tag{88}$$

indeed defines a projective 2-representation of $H_g$ on $\mathcal{S}_g$ in the sense that

$$\big[\Psi_g(h_1,h_2)\otimes\rho_g(h_3)\big] \circ \Psi_g(h_1h_2,h_3) = c_g(h_1,h_2,h_3) \cdot \big[\rho_g(h_1)\otimes\Psi_g(h_2,h_3)\big] \circ \Psi_g(h_1,h_2h_3), \tag{89}$$

where the 3-cocycle $c_g \in Z^3(H_g, U(1))$ depends on the anomaly $\alpha$ and its trivialisation $\psi$. Upon renormalising $\Psi_g \to \gamma_g \cdot \Psi_g$ by an appropriate 2-cochain $\gamma_g \in C^2(H_g, U(1))$, the 3-cocycle $c_g$ can be brought into the canonical form

$$c_g(h_1,h_2,h_3) = \frac{\psi(h_1^g, h_2^g, h_3^g)}{\psi(h_1,h_2,h_3)} \cdot \frac{\alpha(h_1,h_2,h_3,g)\,\alpha(h_1,g,h_2^g,h_3^g)}{\alpha(h_1,h_2,g,h_3^g)\,\alpha(g,h_1^g,h_2^g,h_3^g)}. \tag{90}$$

The interpretation of the projective 2-representation is illustrated in figure 25, where it is shown to represent the compatibility with topological moves of the network of $H$-defects.

We claim that conversely any such projective 2-representation determines a solution to the compatibility constraints for left and right morphisms. The above construction then sets up a bijection between isomorphism classes of simple objects and isomorphism classes of pairs $(g, \Psi_g)$ consisting of

1. A representative $g \in G$ of a double coset $[g] \in H \backslash G / H$.

2. An irreducible projective 2-representation $\Psi_g$ of $H_g$ with 3-cocycle

$$c_g(h_1,h_2,h_3) := \frac{\psi(h_1^g, h_2^g, h_3^g)}{\psi(h_1,h_2,h_3)} \cdot \frac{\alpha(h_1,h_2,h_3,g)\,\alpha(h_1,g,h_2^g,h_3^g)}{\alpha(h_1,h_2,g,h_3^g)\,\alpha(g,h_1^g,h_2^g,h_3^g)}. \tag{91}$$

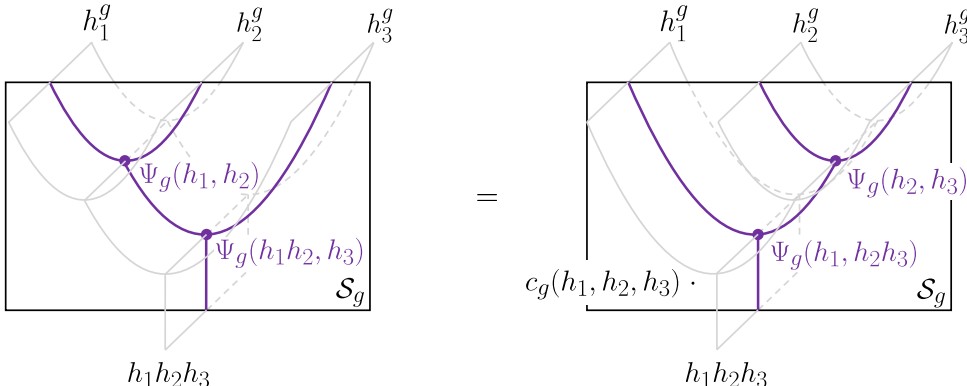

Figure 25

The isomorphism class of a simple object depends on the double coset representative $g$ and the 3-cocycle representative $c_g$ only up to isomorphism.

We can give an alternative description of simple objects using induction of projective 2-representations: In this context, every irreducible projective 2-representation of $H_g$ may be seen as being induced by a 1-dimensional 2-representation of a subgroup of $K \subset H_g$. The latter is completely determined by a choice of 2-cochain $\phi \in C^2(K, U(1))$ satisfying $d\phi = c_g|_K$, which slightly generalises the considerations in [80, 88].

In summary, simple objects are classified by

1. A representative $g \in G$ of a double coset $[g] \in H\backslash G/H$.

2. A subgroup $K \subset H_g$.

3. A 2-cochain $\phi \in C^2(K, U(1))$ satisfying $d\phi = c_g|_K$.

The above description of simple topological lines again allows for an alternative physical interpretation: The topological surface labelled by $g \in G$ in $\mathcal{T}$ is invariant under the action of $H_g \subset H$ and therefore supports a $H_g$ symmetry group. However, due to the bulk 't Hooft anomaly and choice of trivialisation, it has an anomaly $\bar{c}_g \in Z^3(H_g, U(1))$. To define a consistent topological surface when gauging, the anomaly must be cancelled by dressing with an irreducible 2-dimensional TQFT with $H_g$ symmetry and opposite 't Hooft anomaly. This is a projective 2-representation of the above type.

### 3.3.3 1-morphisms

The 1-morphisms in the gauged theory $\mathcal{T}/_\psi H$ are obtained from morphisms in the ungauged theory $\mathcal{T}$ together with compatibility conditions for how they intersect with networks of $H$-defects.

Concretely, given two simple objects $(g, \Psi_g)$ and $(g', \Psi_{g'})$, a 1-morphism between them is obtained from a 1-morphsim $\mathcal{V} : \mathcal{S}_g \to \mathcal{S}'_{g'}$ in $2\mathsf{Vec}^\alpha(G)$. Since this must preserve the grading of the 2-vector spaces $\mathcal{S}_g$ and $\mathcal{S}'_g$, such a morphism can only exist when $g = g'$. This is illustrated in figure 26.

In addition, the 1-morphism $\mathcal{V}$ needs to be equipped with 2-morphisms,

$$\Phi(h) \colon \rho'_g(h) \circ \mathcal{V} \to \mathcal{V} \circ \rho_g(h), \tag{92}$$

in $2\mathsf{Vec}^\alpha(G)$ that describe the intersection of $\mathcal{V}$ with networks of $H$-defects as illustrated in figure 27.

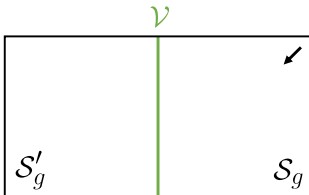

Figure 26

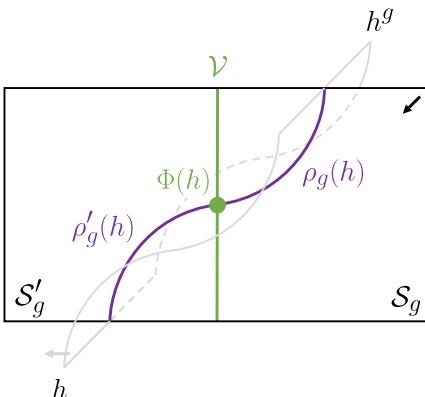

Figure 27

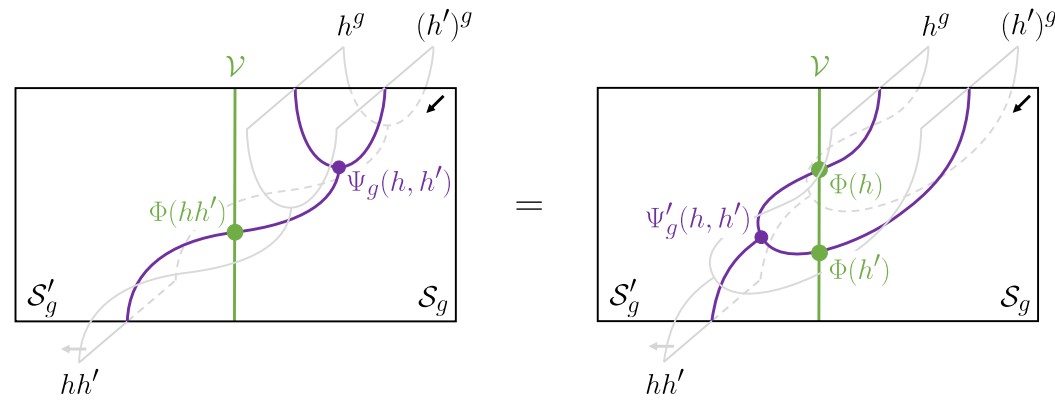

Figure 28

These 2-morphisms must be compatible with topological manipulations of $H$-defects intersecting $\mathcal{S}_g$ and $\mathcal{S}'_g$ in the sense that

$$\Phi(hh') = \frac{\Psi'_g(h,h')}{\Psi_g(h,h')} \cdot \left[\rho'_g(h) \otimes \Phi(h)\right] \circ \left[\Phi(h') \otimes \rho_g(h)\right], \qquad (93)$$

for all $h, h' \in H_g$, which is illustrated in figure 28. This allows us to identify 1-morphisms in $\mathcal{T}/_\psi H$ with graded projective representations (or equivalently 1-intertwiners between 2-representations), which have been studied extensively in Part I [27]. For our purposes, any simple graded projective representation of $H_g$ can be seen as being induced by an ordinary projective representation of a subgroup $K \subset H_g$.

In summary, we obtain a decomposition,

$$\mathsf{C}(G, \alpha \,|\, H, \psi) \;\cong\; \bigoplus_{[g] \in H\backslash G/H} 2\mathrm{Rep}^{c_g}(H_g), \qquad (94)$$

at the level of 2-categories. A generic object will thus be given by a collection of projective 2-representations of subgroups $H_g \subset H$ with 3-cocycles $c_g$ indexed by representatives of double cosets $[g] \in H \backslash G / H$.

Similarly to the two-dimensional case, taking both $H$ and $\psi$ to be trivial reproduces the expected result,

$$C(G, \alpha \,|\, 1) = 2\mathrm{Vec}^\alpha(G), \tag{95}$$

at the level of categories. On the other hand, taking $H = G$ with trivial anomaly gives

$$C(G \,|\, G, \psi) = 2\mathrm{Rep}(G) \tag{96}$$

at the level of categories as anticipated from the discussion in subsection 3.2.

### 3.3.4 Fusion

The fusion of objects is determined by the tensor product of 2-bimodules for the 2-algebra object $A(H, \psi)$ in $2\mathrm{Vec}^\alpha(G)$ associated to $H$ and $\psi$. We will again not present the general formula, but restrict ourselves to some salient features.

Consider two simple objects $S_1$ and $S_2$ supported on double cosets $[g_1]$ and $[g_2]$ respectively. Their fusion should be such that one can consistently insert additional $H$-defects in between them as illustrated in figure 29, and will thus be supported on the decomposition of $[g_1] \cdot [g_2]$ into double cosets.

Analogously to two dimensions, we define the support of a generic object $S$ inside the double coset ring $\mathbb{Z}[H \backslash G / H]$ by

$$\mathrm{sup}(S) := \sum_{[g] \in H \backslash G / H} \dim(\Psi_g) \cdot [g], \tag{97}$$

where we regarded $S$ as a collection $\{\Psi_g\}$ of projective 2-representations indexed by double cosets $[g] \in H \backslash G / H$ as above. The fusion of two objects $S$ and $S'$ must then preserves their support in the sense that

$$\mathrm{sup}(S \otimes S') = \mathrm{sup}(S) * \mathrm{sup}(S'), \tag{98}$$

where $*$ denotes the ring product on $\mathbb{Z}[H \backslash G / H]$.

In this way, the double coset ring again forms the backbone of fusion with respect to the sum decomposition (94). The remaining fusion structure corresponds to decomposing and combining projective 2-representations. We again confine ourselves to specific instances.

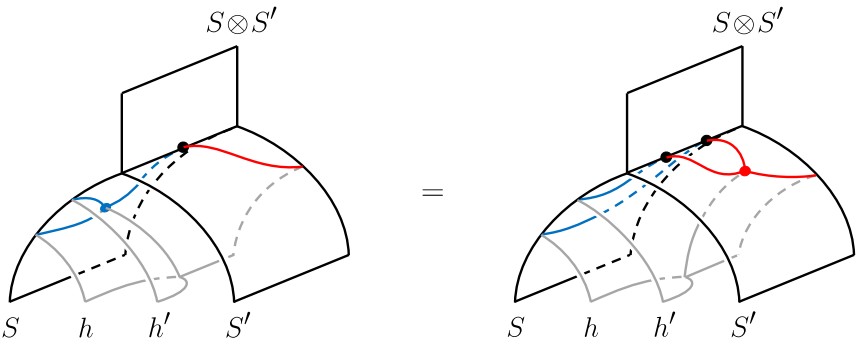

Figure 29

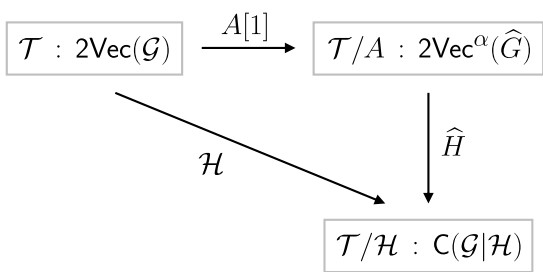

Figure 30

## 3.4 Gauging 2-subgroups

Let us now consider a 3-dimensional theory $\mathcal{T}$ with a finite 2-group symmetry $\mathcal{G}$. This is specified by a 0-form symmetry group $K$, an abelian 1-form symmetry group $A[1]$, a group action $\varphi : K \to \text{Aut}(A)$ and a Postnikov class[5] $[e] \in H^3(K,A)$. In our conventions, specifying local counter terms in the background fields amounts to choosing a representative $e \in Z^3(K,A)$ of the Postnikov class. If the Postnikov class vanishes, one must choose a trivialisation. In this case, shifts of the trivialisation correspond to a choice of symmetry fractionalisation and form a torsor over $H^2(K,A)$.

The system may have an 't Hooft anomaly specified by a class $[\mu] \in H^4(\mathcal{G}, U(1))$ with representative $\mu \in Z^4(\mathcal{G}, U(1))$.[6] The corresponding symmetry category is given by

$$2\text{Vec}^\mu(\mathcal{G}).  \tag{99}$$

Our ambition is to gauge an anomaly-free 2-subgroup $\mathcal{H} \subset \mathcal{G}$. This consists of subgroups $L \subset K$ and $B \subset A$ such that the group action $\varphi : K \to \text{Aut}(A)$ restricts to a group action $\rho : L \to \text{Aut}(B)$ and $e|_L \in Z^3(L,A)$ is valued in $B$. The condition that $\mathcal{H}$ be anomaly-free requires $\mu|_\mathcal{H} = (d\nu)^{-1}$ for some trivialisation $\nu \in C^3(\mathcal{H}, U(1))$. This will result in a 2-group-theoretical fusion 2-category

$$\mathsf{C}(\mathcal{G},\mu \,|\, \mathcal{H}, \nu).  \tag{100}$$

We restrict attention here to cases where the 't Hooft anomaly does not obstruct gauging the whole 1-form symmetry $A[1]$. In this case, $A[1]$ may be gauged first to obtain an ordinary group symmetry $\widehat{G} = \widehat{A} \rtimes_{\widehat{\varphi}} K$ with mixed anomaly, to which we can then apply the machinery from previous subsections. Let us illustrate this procedure by gauging a 2-subgroup $\mathcal{H} \subset \mathcal{G}$ of an anomaly-free 2-group without discrete torsion. The two steps of the gauging procedure are then summarised in 30.

- First, we gauge $A$ without discrete torsion to obtain a theory $\mathcal{T}/A$ with symmetry group $\widehat{G} = \widehat{A} \rtimes_{\widehat{\varphi}} K$. In the presence of a non-trivial Postnikov class, this symmetry has a 't Hooft anomaly $[\alpha] \in H^4(\widehat{G}, U(1))$ with 4-cocycle representative

$$\alpha\big((\chi_1,k_1),(\chi_2,k_2),(\chi_3,k_3),(\chi_4,k_4)\big) = \langle \widehat{\varphi}_{k_1 k_2 k_3}(\chi_4), e(k_1,k_2,k_3) \rangle.  \tag{101}$$

This corresponds to the four-dimensional SPT phase,

$$\int_X \widehat{a} \cup k^*(e),  \tag{102}$$

---

[5]We always assume group the group cohomologies $H^n(G,A)$ to be twisted by the action $\varphi$ of $G$ on $A$.

[6]We use a convenient abuse of notation whereby the singular cohomology of the classifying space of a finite 2-group $\mathcal{G}$ is denoted in a way analogous to finite group cohomology.

in terms of the background fields $\widehat{a} \in H^1(X, \widehat{A})$ and $k : X \to BK$ for the 0-form symmetry $\widehat{G}$. The symmetry category of $\mathcal{T}/A$ is therefore given by

$$C(\mathcal{G}|A) = 2\text{Vec}^\alpha(\widehat{G}).\tag{103}$$

- Next, we note that we can relate the 2-subgroup $\mathcal{H} \subset \mathcal{G}$ in $\mathcal{T}$ to a corresponding ordinary subgroup $\widehat{H} \subset \widehat{G}$ in $\mathcal{T}/A$ as follows:

  ○ Given a 2-subgroup $\mathcal{H} = (L, B)$ of $\mathcal{G}$, there is an associated short exact sequence for the 1-form parts

  $$1 \to B \xrightarrow{\iota} A \xrightarrow{\pi} C := A/B \to 1,\tag{104}$$

  which can be dualised to obtain a short exact sequence

  $$1 \to \widehat{C} \xrightarrow{\widehat{\pi}} \widehat{A} \xrightarrow{\widehat{\iota}} \widehat{B} \to 1,\tag{105}$$

  for the corresponding Pontryagin dual groups. Let now $l \in L$ and $\chi \in \widehat{C}$. Using that by assumption the group action $\varphi$ restricts to a group action of $L$ on $B$, it is then straighforward to check that

  $$\langle \widehat{\iota}(l \triangleright \widehat{\pi}(\chi)), b \rangle = \langle \chi, (\pi \circ \iota)(l^{-1} \triangleright b) \rangle \equiv 1,\tag{106}$$

  for all $b \in B$, which implies that $l \triangleright \widehat{\pi}(\chi) \in \ker(\widehat{\iota}) = \text{im}(\widehat{\pi})$. Thus, the Pontryagin dual action $\widehat{\varphi}$ restricts to an action of $L$ on $\widehat{\pi}(\widehat{C}) \subset \widehat{A}$, which allows us to define a subgroup $\widehat{H} := \widehat{\pi}(\widehat{C}) \rtimes_{\widehat{\varphi}} L$ of $\widehat{G}$. Furthermore, since $e|_L$ is valued in $B$ by assumption, we have that

  $$\langle \widehat{\pi}(\chi), e(l_1, l_2) \rangle = \langle \chi, (\pi \circ \iota)(e(l_1, l_2)) \rangle \equiv 1,\tag{107}$$

  for all $\chi \in \widehat{C}$ and $l_1, l_2 \in L$, which is equivalent to saying that the anomaly $\alpha$ from (101) becomes trivial upon restriction to $\widehat{H} \subset \widehat{G}$.

  ○ Conversely, running through the above arguments backwards shows that any subgroup $\widehat{H} \subset \widehat{G}$ with $\alpha|_{\widehat{H}} = 1$ uniquely determines a 2-subgroup $\mathcal{H}$ of $\mathcal{G}$.

  In summary, there is a 1-1 correspondence between 2-subgroups $\mathcal{H} \subset \mathcal{G}$ and subgroups $\widehat{H} \subset \widehat{G}$ with $\alpha|_{\widehat{H}} = 1$ given by

  $$\mathcal{H} = (L, B) \quad \longleftrightarrow \quad \widehat{H} = \widehat{A/B} \rtimes L.\tag{108}$$

  Gauging the 2-subgroup $\mathcal{H}$ in $\mathcal{T}$ can thus be achieved by gauging the subgroup $\widehat{H}$ in $\mathcal{T}/A$ using the machinery from section 3.3. The symmetry category of $\mathcal{T}/\mathcal{H}$ is therefore given by

  $$C(\mathcal{G}|\mathcal{H}) = C(\widehat{G}, \alpha|\widehat{H}).\tag{109}$$

## 3.5 Case study I

Let us now consider the case where we gauge the whole 2-group symmetry $\mathcal{G}$ of $\mathcal{T}$. This must result in the symmetry category $2\text{Rep}(\mathcal{G})$, but it is illuminating to reproduce this result by gauging in steps: We first gauge the entire 1-form symmetry $A[1]$ to obtain a theory $\mathcal{T}/A$ with symmetry group $\widehat{G} = \widehat{A} \rtimes_{\widehat{\varphi}} K$ and mixed 't Hooft anomaly $\alpha$ and subsequently gauge the remaining 0-form symmetry $K \subset \widehat{G}$ as shown in figure 31. This generalises the computation that was done for split 2-groups in Part I [27].

In order to describe simple objects in $\mathcal{T}/\mathcal{G}$, we first note that double $K$-cosets in $\widehat{G}$ are in 1-1 correspondence with $K$-orbits in $\widehat{A}$. Let us choose a representative $\chi \in \widehat{A}$ of a $K$-orbit $\mathcal{O}(\chi)$

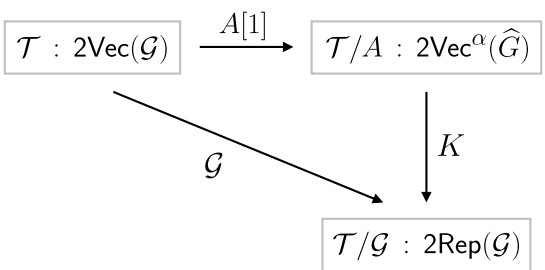

Figure 31

with stabiliser $\text{Stab}(\chi) = K \cap {}^{\chi}K$. Then, the 3-cocycle $c_{\chi}$ from (90) with $\alpha$ as in (101) and $\psi = 1$ reduces to

$$c_{\chi}(k_1, k_2, k_3) = \langle \widehat{\varphi}_{k_1 k_2 k_3}(\chi), e(k_1, k_2, k_3) \rangle . \tag{110}$$

The simple objects are therefore labelled by triples consisting of

1. a $K$-orbit $\mathcal{O}(\chi) \subset \widehat{A}$ with representative $\chi$,

2. a subgroup $L \subset \text{Stab}(\chi)$ of its stabiliser,

3. a 2-cochain $\phi \in C^2(L, U(1))$ satisfying $d\phi = \langle \chi, e|_L \rangle$.

This is equivalent to the data of a finite-dimensional 2-representation of the 2-group $\mathcal{G}$ [81] and reduces to the construction of simple objects that was presented in Part I [27] for the case of a split 2-group with $[e] = 0$. In summary, gauging the 2-group $\mathcal{G}$ in steps reproduces the expected result

$$C(\mathcal{G} | \mathcal{G}) = 2\text{Rep}(\mathcal{G}). \tag{111}$$

### 3.5.1 Example: $\mathcal{G} = \mathbb{Z}_2[1] \times \mathbb{Z}_2$

Consider the case where $K = \mathbb{Z}_2$ and $A[1] = \mathbb{Z}_2$. We denote the generators of $A$ and $K$ by $x$ and $y$, respectively. There are two possible 2-group structures[7] corresponding to the two possible Postnikov classes

$$[e] \in H^3(\mathbb{Z}_2, \mathbb{Z}_2) = \mathbb{Z}_2, \tag{112}$$

with normalised cocycle representatives $e(y, y, y) = 1$ and $e(y, y, y) = x$ respectively. We call the corresponding 2-groups split and non-split respectively. The simple objects after gauging can then be constructed as follows:

- For the split 2-group, there are no non-trivial 2-cocycles $\phi$ since $H^2(\mathbb{Z}_2, U(1)) = 0$. The simple objects are therefore completely determined by a choice of character $\chi \in \widehat{A}$ and subgroup $L \subset \mathbb{Z}_2$ of the stabiliser. We thus have four simple objects

$$
\begin{array}{c|ccc}
 & \chi & L & \phi \\
\hline
1 & 1 & \mathbb{Z}_2 & 1 \\
X & 1 & \{1\} & 1 \\
V & \widehat{x} & \mathbb{Z}_2 & 1 \\
X' & \widehat{x} & \{1\} & 1 \\
\end{array}
\tag{113}
$$

---

[7]There is no choice of symmetry fractionalisation since $H^2(\mathbb{Z}_2, \mathbb{Z}_2) = 0$.

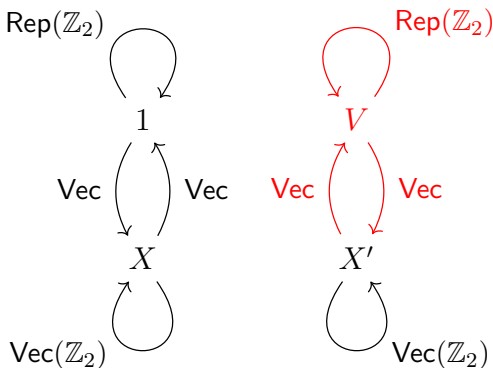

Figure 32

whose fusion rules can be determined to be

$$
\begin{aligned}
V \otimes V &= 1\,, \\
V \otimes X &= X'\,, \\
X \otimes X &= X' \otimes X' = 2X\,, \\
X \otimes X' &= 2X'\,.
\end{aligned}
\tag{114}
$$

Physically, $V$ is the generator of the dual 0-form symmetry that results from gauging $A = \mathbb{Z}_2$ and generates a subcategory $2\mathrm{Rep}(\mathbb{Z}_2[1]) \cong 2\mathrm{Vec}(\mathbb{Z}_2)$. On the other hand, $X$ is the condensation defect for topological Wilson lines that result from gauging $K = \mathbb{Z}_2$ and generates a subcategory $2\mathrm{Rep}(\mathbb{Z}_2) \cong 2\mathrm{Vec}(\mathbb{Z}_2[1])$. The total symmetry category is given by

$$
2\mathrm{Rep}(\mathbb{Z}_2[1] \times \mathbb{Z}_2) \cong 2\mathrm{Vec}(\mathbb{Z}_2 \times \mathbb{Z}_2[1])\,.
\tag{115}
$$

Its simple objects and 1-morphism spaces are illustrated in figure 32.

- For the non-split 2-group, the condition $d\phi = \langle \chi, e|_L \rangle$ becomes non-trivial only when $\chi = \hat{x}$ and $L = \mathbb{Z}_2$. In this case, since no such normalised 2-cochain $\phi$ exists, the corresponding defect $V$ is no longer present in the spectrum. This is indicated by the red colouring of the defect $V$ and its attached 1-morphism spaces in figure 32. The remaining 1-morphisms and fusion rules are the same as before.

### 3.5.2  Example: $\mathcal{G} = \mathbb{Z}_4[1] \rtimes \mathbb{Z}_2$

As another example, suppose now $K = \mathbb{Z}_2$ and $A[1] = \mathbb{Z}_4$. We denote the generators of $A$ and $K$ by $x$ and $y$ again and assume a non-trivial group action with homomorphism fixed by $\varphi_y(x) = x^3$. There are again two possible 2-groups $\mathcal{G}$ corresponding to the two possible Postnikov classes

$$
[e] \in H^3(\mathbb{Z}_2, \mathbb{Z}_4) = \mathbb{Z}_2\,,
\tag{116}
$$

with normalised representatives $e(y, y, y) = 1$ and $e(y, y, y) = x$ (which are cohomologous to $e(y, y, y) = x^2$ and $e(y, y, y) = x^3$ respectively). The simple objects after gauging can then be constructed as follows:

- For the split 2-group, there are again no non-trivial 2-cocycles $\phi$ so that the simple objects are completely determined by a choice of orbit representative $\chi \in \widehat{A}$ and subgroup

$L$ of the stabiliser. There are now five simple objects

$$
\begin{array}{c|ccc}
 & \chi & L & \phi \\
\hline
1 & 1 & \mathbb{Z}_2 & 1 \\
X & 1 & \{1\} & 1 \\
D & \widehat{x} & \{1\} & 1 \\
V & \widehat{x}^2 & \mathbb{Z}_2 & 1 \\
X' & \widehat{x}^2 & \{1\} & 1
\end{array}
\tag{117}
$$

whose fusion rules are as in (114) with additional relations

$$
\begin{aligned}
V \otimes D &= D, \\
X \otimes D &= 2D, \\
D \otimes D &= X \oplus X'.
\end{aligned}
\tag{118}
$$

Note that $X, X'$ are again condensation defects for the topological Wilson lines obtained from gauging $K = \mathbb{Z}_2$. The simple objects and 1-morphism spaces in the resulting symmetry category $2\mathrm{Rep}(\mathbb{Z}_4[1] \rtimes \mathbb{Z}_2)$ are illustrated in figure 33.

- For the non-split 2-group, the condition $d\phi = \langle \chi, e|_L \rangle$ is non-trivial only when $\chi = \widehat{x}^2$ and $L = \mathbb{Z}_2$. In this case, since no such normalised 2-cochain $\phi$ exists, the corresponding defect $V$ is no longer present in the spectrum. This is indicated by the red colouring of the defect $V$ and its attached morphism spaces in figure 33. The remaining 1-morphisms and fusion rules are the same as before.

Finally, we note that replacing $\mathbb{Z}_4$ by $D_4 = \mathbb{Z}_2 \times \mathbb{Z}_2$ with $\mathbb{Z}_2$-action exchanging the two factors leads to the same spectra of simple objects and equivalent symmetry categories

$$
2\mathrm{Rep}(\mathbb{Z}_4[1] \rtimes \mathbb{Z}_2) \cong 2\mathrm{Rep}(D_4[1] \rtimes \mathbb{Z}_2),
\tag{119}
$$

despite the fact that $\mathbb{Z}_4[1] \rtimes \mathbb{Z}_2$ and $D_4[1] \rtimes \mathbb{Z}_2$ are distinct 2-groups.

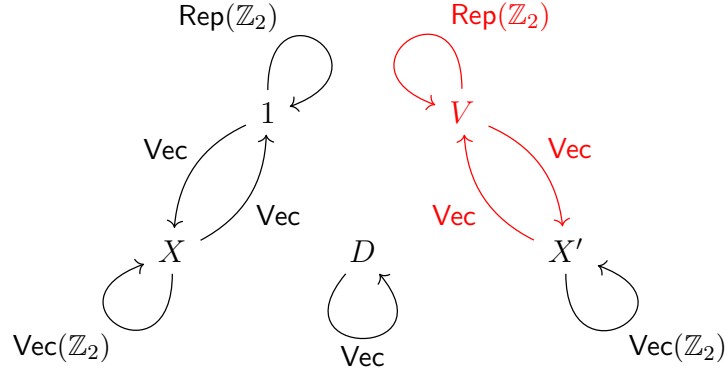

Figure 33

### 3.6 Case study II

Let us consider a theory $\mathcal{T}$ with anomaly free symmetry $G = D_8$ and systematically gauge all possible subgroups $H \subset G$ with discrete torsion. The possible choices corresponds to gapped boundary conditions for 4-dimensional Dijkgraaf-Witten theory for $D_8$ with trivial topological action, which acts as symmetry TFT for the resulting class of symmetries.

Our primary example will be 3-dimensional Yang-Mills theory with gauge group $PSO(N)$ with $N$ even, whose magnetic and charge conjugation symmetries combine to form $D_8$. Gauging subgroups of this symmetry will provide a systematic analysis of the fusion 2-category symmetries of various global forms of gauge theories based on the Lie algebra $\mathfrak{so}(N)$, including those with disconnected gauge groups and discrete theta angles.

If $N = 4k + 2$, we introduce standard generators $r$ and $s$ and present $D_8$ as

$$D_8 = \langle r, s \, | \, r^4 = s^2 = 1, \, srs^{-1} = r^{-1} \rangle, \tag{120}$$

which identifies the symmetry group with the semi-direct product $\mathbb{Z}_4 \rtimes \mathbb{Z}_2$. In this formulation, $\mathbb{Z}_4$ corresponds to the magnetic symmetry $\pi_1(PSO(N))^\vee$ and $\mathbb{Z}_2$ to the charge conjugation symmetry $\mathrm{Out}(PSO(N))$.

If $N = 4k$, we introduce generators $a = rs$ and $b = sr$ and present $D_8$ as

$$D_8 = \langle a, b, s \, | \, a^2 = b^2 = s^2 = 1, \, ab = ba, \, sas^{-1} = b \rangle, \tag{121}$$

which identifies the symmetry group with the semi-direct product $(\mathbb{Z}_2 \times \mathbb{Z}_2) \rtimes \mathbb{Z}_2$. In this formulation, $\mathbb{Z}_2 \times \mathbb{Z}_2$ corresponds to the magnetic symmetry $\pi_1(PSO(N))^\vee$ and $\mathbb{Z}_2$ to the charge conjugation symmetry $\mathrm{Out}(PSO(N))$. For simplicity, we will focus on this example in what follows.

We remind the reader that the subgroup and automorphism structure of $D_8$ is summarised in figure 14. We now consider the symmetry categories that result from gauging subgroups with discrete torsion, beginning with subgroups of the smallest order and working upwards in figure 14.

#### 3.6.1 Order two subgroups

We begin by gauging the order 2 subgroups isomorphic to $H \cong \mathbb{Z}_2$. In this case, it is possible to gauge with discrete torsion corresponding to the non-trivial class in $H^3(\mathbb{Z}_2, U(1)) \cong \mathbb{Z}_2$, which may be represented by adding a counter term of the form

$$\frac{1}{2} \int \mathsf{a} \cup \mathsf{a} \cup \mathsf{a}. \tag{122}$$

There are 5 order two subgroups forming 3 conjugacy classes, two of which are related by an outer automorphism. There are therefore only two substantive cases to consider:

- The center $H = \langle r^2 \rangle \cong \mathbb{Z}_2$ of $D_8$ forms a non-split extension

$$1 \to \mathbb{Z}_2 \to D_8 \to D_4 \to 1, \tag{123}$$

  with non-trivial extension class $[e] \in H^2(D_4, \mathbb{Z}_2)$. The extension class may be represented by the two-dimensional SPT phase,

$$\frac{1}{2} \int \mathsf{a}_1 \cup \mathsf{a}_2, \tag{124}$$

  in terms of the background fields $\mathsf{a}_1, \mathsf{a}_2 \in H^1(X, \mathbb{Z}_2)$ for the $D_4$ symmetry. Gauging the center will result in an $SO(N)$ gauge theory. However, the global structure and symmetry

category will depend on the choice of discrete torsion. we denote the choice of discrete torsion by $\phi \in \mathbb{Z}_2$ and the resulting global form may be expressed as

$$SO(N)_\phi = \frac{SO(N) \times \mathcal{D}(\mathbb{Z}_2)_\phi}{\mathbb{Z}_2[1]}, \tag{125}$$

where the quotient means gauging the diagonal $\mathbb{Z}_2$ 1-form symmetry [70,71]. Here and in the following we denote by $\mathcal{D}(H)_\phi$ denotes the 3-dimensional Dijkgraaf-Witten theory associated to $\phi \in H^3(H, U(1))$.

- In the absence of discrete torsion ($\phi = 0$), gauging $H \cong \mathbb{Z}_2$ results in a split 2-group symmetry $\mathbb{Z}_2[1] \times D_4$ with 't Hooft anomaly determined by the extension class $[e]$, which can be represented by the cubic SPT phase

$$\frac{1}{2} \int_X \widehat{\mathsf{a}} \cup \mathsf{a}_1 \cup \mathsf{a}_2, \tag{126}$$

  where $\widehat{\mathsf{a}} \in H^2(X, \mathbb{Z}_2)$ denotes the background for the $\mathbb{Z}_2[1]$ symmetry. The corresponding global form is the plain $SO(N)_0$ gauge theory.

- Now consider gauging with non-trivial discrete torsion ($\phi = 1$). This can be understood via the Lyndon-Hochschild-Serre spectral sequence associated to the short exact sequence of groups (123) in a manner analogous to section 2.4 and appendix A. In this instance, the first obstruction vanishes and the second obstruction corresponding to the differential,

$$d_3^{0,3}: H^3(\mathbb{Z}_2, U(1)) \to H^3(D_4, U(1)), \tag{127}$$

  sends the discrete torsion to an additional contribution to the 't Hooft anomaly represented by the SPT phase

$$\frac{1}{2} \int_X \mathcal{P}(\mathsf{a}_1 \cup \mathsf{a}_2), \tag{128}$$

  where $\mathcal{P}: H^2(-, \mathbb{Z}_2) \to H^4(-, \mathbb{Z}_4)$ is the Pontryagin square operation. The spectral sequence computation is performed explicitly in [72]. The same computation is performed in [71] using an explicit Chern-Simons theory representation. This corresponds to a distinct global form $SO(N)_1$.

In summary, gauging the centre $H = \langle r^2 \rangle$ with discrete torsion $\phi \in \mathbb{Z}_2$ leads to the global form $SO(N)_\phi$ with symmetry category

$$\mathsf{C}(D_8 \,|\, \langle r^2 \rangle, \phi) = 2\mathsf{Vec}^{\alpha_\phi}(\mathbb{Z}_1[1] \times D_4), \tag{129}$$

where the anomaly $\alpha_\phi$ is represented by the SPT phase

$$\frac{1}{2} \int_X \widehat{\mathsf{a}} \cup \mathsf{a}_1 \cup \mathsf{a}_2 + \frac{\phi}{2} \int_X \mathcal{P}(\mathsf{a}_1 \cup \mathsf{a}_2). \tag{130}$$

The result of adding discrete torsion is thus to shift 't Hooft anomaly in the resulting symmetry category.

- Now consider the two non-normal subgroups $H = \langle s \rangle, \langle r^2 s \rangle \cong \mathbb{Z}_2$, which are related to each other by conjugation. For concreteness, consider gauging charge conjugation $H = \langle s \rangle$. Gauging this subgroup results in a $PO(N)$ gauge theory. However, the specific global form and symmetry category will depend on the choice of discrete torsion when gauging.

○ First consider the case without discrete torsion. The simple objects can be determined as follows. There are three double cosets $[1], [r], [r^2]$ with stabilisers $H, 1, H$ respectively and double coset ring

$$[r] * [r] = [1] + [r^2], \qquad [r] * [r^2] = [r], \qquad [r^2] * [r^2] = [1]. \qquad (131)$$

There are therefore 5 simple objects corresponding to the following pairs of double cosets and irreducible representations

$$\begin{aligned} 1 &= ([1], 1), & X &= ([1], \omega), & D &= ([r], 1), \\ V &= ([r^2], 1), & X' &= ([r^2], \omega), & & \end{aligned} \qquad (132)$$

where $\omega$ denotes the non-trivial irreducible 2-representation (or condensation defect) of $\mathbb{Z}_2$. The fusion ring takes the following form:

$$\begin{aligned} V \otimes V &= 1, & D \otimes D &= X \oplus X', \\ V \otimes D &= D, & X \otimes D &= D \oplus D, \\ V \otimes X &= X', & X \otimes X &= 2X. \end{aligned} \qquad (133)$$

The symmetry category is identified with

$$C(D_8 \,|\, \langle s \rangle) = 2\mathrm{Rep}(\mathbb{Z}_4[1] \rtimes \mathbb{Z}_2). \qquad (134)$$

To understand this result, note that one may first gauge the subgroup $\langle r \rangle \cong \mathbb{Z}_4$ to obtain a dual 2-group symmetry $\mathbb{Z}_4[1] \rtimes \mathbb{Z}_2$. Then, gauging the entire 2-group symmetry reproduces the $PO(N)$ theory and symmetry category $2\mathrm{Rep}(\mathbb{Z}_4[1] \rtimes \mathbb{Z}_2)$. An analogous statement holds if we replace $\langle r \rangle \cong \mathbb{Z}_4$ by $\langle rs, r^3 s \rangle \cong D_4$, making use of the fact that

$$2\mathrm{Rep}(\mathbb{Z}_4[1] \rtimes \mathbb{Z}_2) \cong 2\mathrm{Rep}(D_4[1] \rtimes \mathbb{Z}_2). \qquad (135)$$

The above results are compatible with the fusion rules derived in [33]. The non-invertible defect $\mathcal{N}_1$ there is identified with the 2-dimensional 2-representation $D$, while the symmetry defect $W$ is identified with the 1-dimensional 2-representation $V$, and $X$ is the condensation.

○ Adding a non-trivial discrete torsion when gauging results in a $PO(N)$ gauge theory with a discrete theta angle

$$\frac{1}{2} \int w_1 \cup w_1 \cup w_1, \qquad (136)$$

where $w_1$ denotes the first Stiefel-Whitney class obstructing the restriction of a $PO(N)$ bundle to a $PSO(N)$ bundle [70]. Since $H = \langle s \rangle$ is not a normal subgroup of $D_8$, we cannot utilise a spectral sequence construction to determine the symmetry category.

• Now consider the two non-normal subgroups $H = \langle rs \rangle, \langle r^3 s \rangle \cong \mathbb{Z}_2$, which are related to reach other by conjugation. Gauging these subgroups results in $Ss(N)$ and $Sc(N)$ gauge theories respectively. The two subgroups are related to those considered in the previous bullet point by an outer automorphism and therefore the construction of the symmetry category is identical to above.

### 3.6.2 Order four subgroups

Recall that there are three order four subgroups, all of which are normal: one is isomorphic to $\mathbb{Z}_4$ and invariant under the outer automorphism and the remaining two are isomorphic to $D_4$ and exchanged by the outer automorphism. In both cases there is the opportunity to add discrete torsion since

$$
\begin{aligned}
H^3(\mathbb{Z}_4, U(1)) &= \mathbb{Z}_4, \\
H^3(D_4, U(1)) &= \mathbb{Z}_2^3.
\end{aligned}
\tag{137}
$$

We consider the resulting symmetry 2-categories in turn:

- Let us first consider the normal subgroup $H = \langle r^2, s \rangle \cong D_4$. Gauging this subgroup results in a 2-group symmetry $D_4[1] \rtimes \mathbb{Z}_2$. Since $H$ forms a split short exact sequence with $D_8$, there are no obstructions and discrete torsion acts on the resulting symmetry 2-category by an auto-equivalence. In summary,

$$
\mathsf{C}(D_8 \,|\, D_4, \phi) = 2\mathrm{Vec}(D_4[1] \rtimes \mathbb{Z}_2).
\tag{138}
$$

  In our running example, this results in an $O(N)^0$ gauge theory and the effect of adding discrete torsion is to alternate between different global forms. On the one hand, introducing discrete torsion for the $\mathbb{Z}_2$ subgroup $\langle s \rangle \subset H$ corresponds to adding a discrete theta angle

$$
\frac{1}{2} \int w_1 \cup w_1 \cup w_1,
\tag{139}
$$

  where $w_1$ now denotes the first Stiefel-Whitney class obstructing the lift of an $O(N)$-bundle to an $SO(N)$-bundle. On the other hand, introducing discrete torsion for the $\mathbb{Z}_2$ subgroup $\langle r^2 \rangle \subset H$ corresponds to the global form

$$
O(N)_\phi = \frac{O(N) \times \mathcal{D}(\mathbb{Z}_2)_\phi}{\mathbb{Z}_2[1]}.
\tag{140}
$$

  There is one further generator of discrete torsion and 8 possible global forms given the $\mathbb{Z}_2^3$ classification in (137). Our analysis shows that all of these global forms share the same symmetry category up to equivalence.

- The remaining normal $D_4$ subgroup $H = \langle r^2, rs \rangle$ is related to the one above by an outer automorphism and therefore leads to an identical analysis for the symmetry categories. They correspond to $Spin(N)$ gauge theories with discrete torsion resulting in different global forms

$$
Spin(N)_\phi = \frac{Spin(N) \times \mathcal{D}(D_4)_\phi}{D_4[1]},
\tag{141}
$$

  where $\phi \in H^3(D_4, U(1)) \cong \mathbb{Z}_2^3$.

- Finally, consider the normal subgroup $H = \langle r \rangle \cong \mathbb{Z}_4$. Gauging this subgroup leads to a split 2-group symmetry $\mathbb{Z}_4[1] \rtimes \mathbb{Z}_2$. Since $H$ forms a split short exact sequence with $D_8$, there are no obstructions and discrete torsion $[\phi] \in H^3(\mathbb{Z}_4, U(1))$ acts on the resulting symmetry 2-category by an auto-equivalence. In summary,

$$
\mathsf{C}(D_8 \,|\, \mathbb{Z}_4, \phi) = 2\mathrm{Vec}(\mathbb{Z}_4[1] \rtimes \mathbb{Z}_2).
\tag{142}
$$

  In our running example, gauging $H = \mathbb{Z}_4$ leads to a $O(N)^1$ gauge theory, where the superscript 1 denotes the presence of the discrete theta angle

$$
\frac{1}{2} \int w_1 \cup w_2.
\tag{143}
$$

Here, $w_1$ and $w_2$ are the first and second Stiefel-Whitney class of $O(N)$-bundles. One way to understand this interpretation is to gauge in steps. Recall that first gauging the central subgroup $\langle r^2 \rangle$ reproduces an $SO(N)$ gauge theory. The remaining 0-form symmetries correspond to the magnetic symmetry $\langle rs \rangle \cong \mathbb{Z}_2$ and charge conjugation $\langle s \rangle \cong \mathbb{Z}_2$. Subsequently gauging the diagonal combination of these symmetries, which in our notation corresponds to gauging $\langle r \rangle$, reproduces the $O(N)^1$ theory [70].

The effect of adding discrete torsion $\phi \in H^3(\mathbb{Z}_4, U(1)) = \mathbb{Z}_4$ corresponds to different global forms of an $O(N)^1$ gauge theory

$$O(N)^1_\phi = \frac{O(N)^1 \times \mathcal{D}(\mathbb{Z}_4)_\phi}{\mathbb{Z}_2[1]}. \tag{144}$$

Our analysis shows that these global forms share the same symmetry 2-category up to equivalence.

### 3.6.3 Gauging the whole group

Finally, we may gauge the entire symmetry group $H = D_8$ together with discrete torsion

$$[\phi] \in H^3(D_8, U(1)) \cong \mathbb{Z}_2 \times \mathbb{Z}_2 \times \mathbb{Z}_4. \tag{145}$$

The resulting symmetry 2-category is given by $\mathsf{C}(D_8 \,|\, D_8, \phi) = 2\mathrm{Rep}(D_8)$.

In our running example, this corresponds to a $Pin^\pm(N)$ gauge theory, where the choice of $\pm$ and specific global form depends on the choice of discrete torsion. In order to enumerate the possibilities and understand their physical interpretation, it is convenient to use as an organisational tool the Lyndon-Hochschild-Serre spectral sequence to enumerate possible discrete torsion. This does not necessarily reproduce the group structure on (145), but it is a convenient way to identify specific discrete torsion elements and their physical interpretation. There are many ways to do this and we provide two illustrative examples below.

Let us first consider the split short exact sequence

$$1 \to D_4 \to D_8 \to \mathbb{Z}_2 \to 1, \tag{146}$$

that is associated to the semi-direct product structure $D_8 \cong D_4 \rtimes \mathbb{Z}_2$. One discrete torsion element of interest arises from the term

$$E_2^{3,0} = H^3(\mathbb{Z}_2, U(1)) \cong \mathbb{Z}_2. \tag{147}$$

This corresponds to gauging the $\mathbb{Z}_2$ charge conjugation symmetry of $Spin(N)$ gauge theory with discrete torsion and reproduces the $Pin^+(N)$ gauge theory with discrete theta angle

$$\frac{1}{2} \int w_1 \cup w_1 \cup w_1, \tag{148}$$

where $w_1$ denotes the first Stiefel-Whitney class that obstructs lifting a $Pin^+(N)$-bundle to a $Spin(N)$-bundle.

Now consider instead the short exact sequence

$$1 \to \mathbb{Z}_4 \to D_8 \to \mathbb{Z}_2 \to 1, \tag{149}$$

associated to the semi-direct product structure $D_8 \cong \mathbb{Z}_4 \rtimes \mathbb{Z}_2$. We now consider the discrete torsion element arising from the term

$$E_2^{2,1} = H^2(\mathbb{Z}_2, \mathbb{Z}_4) \cong \mathbb{Z}_2, \tag{150}$$

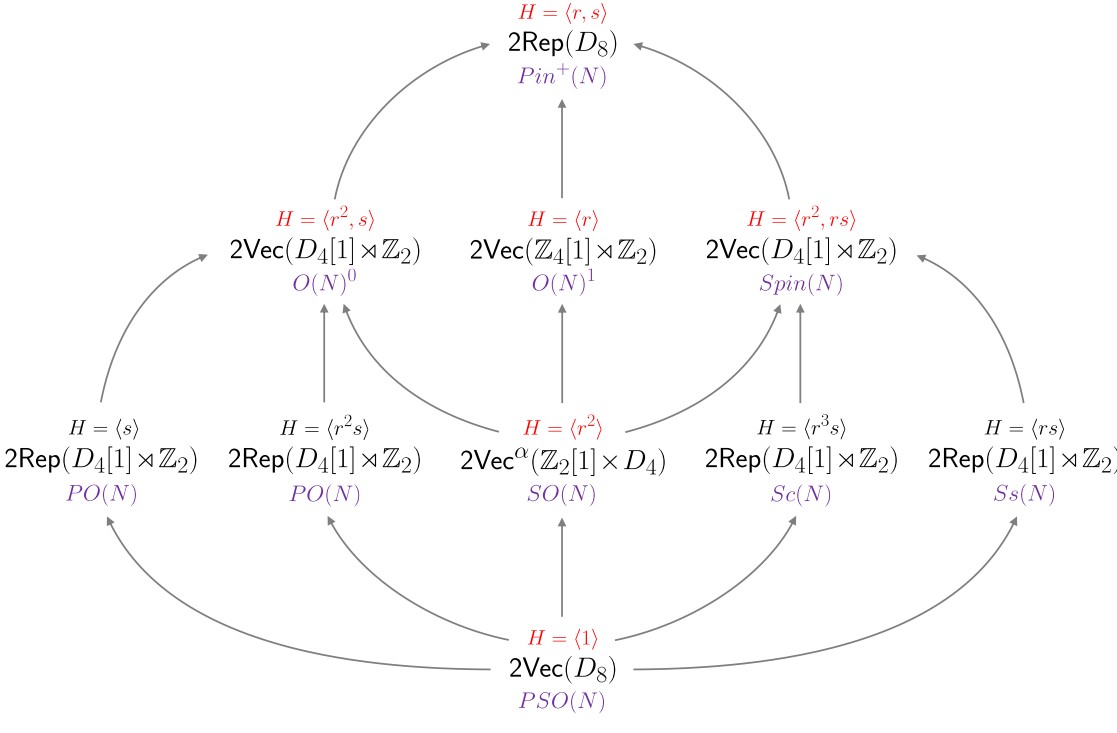

Figure 34

where $\mathbb{Z}_4$ is understood as a non-trivial $\mathbb{Z}_2$-module. This corresponds to first gauging the $\mathbb{Z}_4$ symmetry of the $PSO(N)$ theory with a local counter term

$$\frac{1}{4} \int \mathsf{k}^*(\phi) \cup \mathsf{a} \,, \tag{151}$$

where $\mathsf{a}$ is the dynamical $\mathbb{Z}_4$ background and $\mathsf{k}$ denotes the background for the remaining $\mathbb{Z}_2$ symmetry. The result is a $O(N)^1$ gauge theory where the background $\hat{\mathsf{a}}$ for the emergent $\mathbb{Z}_4[1]$ symmetry is shifted by

$$\hat{\mathsf{a}} \to \hat{\mathsf{a}} + \mathsf{k}^*(\phi) \,. \tag{152}$$

If $\phi$ is non-trivial, this corresponds to adding a non-trivial symmetry fractionalisation. Subsequently gauging the remaining $\mathbb{Z}_2$ symmetry then results in a $Pin^-(N)$ gauge theory [70].

There are many compatibility checks as order four subgroups may also be gauged by gauging order two subgroups in steps via composition of arrows in figure 14. The above results are summarised in figure 34, in which we have omitted the outcomes of gauging with discrete torsion for brevity.

## 4  Four dimensions

In this section, we consider gauging 3-subgroups of finite 3-groups in four dimensions. One expects on general grounds (and under mild assumptions) that the associated symmetry categories are fusion 3-categories, which are expected to be even richer and more intricate than fusion 2-categories. As the mathematical literature on the topic is less developed, we do not wish to be systematic but to provide some general considerations and leverage the knowledge we have acquired in lower dimensions.

An intuitive reason for the increase in richness is that topological lines on a three-dimensional topological defect $\Omega$ may braid as illustrated in figure 35. This is reflected in

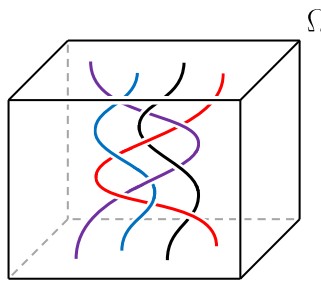

Figure 35

an increase in richness of 3-dimensional TQFTs compared with one and two dimensions. A corresponding observation is that while topological order in one and two dimensions is well described by SPT phases, there are also SET phases in three dimensions [82–84].

From an algebraic perspective, we now work with 3Vec, whose objects are 3-dimensional framed fully-extended TQFTs of Turaev-Viro type. As a result, our formalism does not capture all possible topological defects in four dimensions that have been considered in the literature, such as the topological defects constructed in [33]. Incorporating said defects would require a further enlargement of 3Vec to include 3-dimensional TQFTs that are not of Turaev-Viro type. A proper treatment of such an enlargement is beyond the scope of this paper and is left to future work.

The structure of 3Vec considered in what follows underpins constructions in this section as well as 3-representation theory more broadly. Let us compare the situation with sections 2 and 3:

- Objects of Vec are finite-dimensional vector spaces, $\mathbb{C}^n$

- Objects of 2Vec are finite-dimensional 2-vector spaces, $\mathsf{Vec}^n$.

- Objects 3Vec include finite-dimensional 3-vector spaces of the form $2\mathsf{Vec}^n$. However, more generally it contains objects $\mathsf{Mod}(Z(\mathsf{C}))$ for some multi-fusion category C, corresponding to 3d TQFTs obtained by the Turaev-Viro construction. This includes the former by taking $\mathsf{C} = \mathsf{Vec}^n$.

The additional 3-vector spaces beyond $2\mathsf{Vec}^n$ may serve as the receptacle for new types of 3-representations and projective 3-representations that involve distinct new phenomena compared with 1- and 2-representations.

This additional structure permeates the investigation of non-invertible symmetries in $D = 4$. One way it manifests is in the appearance of TQFT valued coefficients in fusion rules of non-invertible symmetries in four dimensions [32, 33, 43]. If we consider the fusion of a topological surface $\mathcal{S}$ with a decoupled TQFT $\mathcal{A}$ corresponding to some object in 3Vec. If $\mathcal{A} = 2\mathsf{Vec}^n$, this produces a direct sum

$$\mathcal{A} \otimes \mathcal{S} = \mathcal{S} \oplus \cdots \oplus \mathcal{S} = n \cdot \mathcal{S}, \tag{153}$$

much as in two and three dimensions. However, if $\mathcal{A}$ supports topological lines that braid non-trivially, then $\mathcal{A} \otimes \mathcal{S}$ does not admit such a decomposition. Such contributions arise in the fusion rules of non-invertible symmetries in four dimensions and have been interpreted as TQFT-valued coefficients.

It also manifests when gauging 1-form symmetries with 't Hooft anomalies, where the dressing by an anomalous TQFT is reformulated in terms of projective 3-representations of

the 1-form symmetry. Higher representation theory provides a tool to systematise such examples and many computations boil down to higher analogues of classical constructions in the representation theory of finite groups.

## 4.1 Preliminaries

Let us consider a theory $\mathcal{T}$ with anomaly-free finite group symmetry $G$. The symmetry 3-category $3\mathrm{Vec}(G)$ contains simple objects labelled by group elements $g \in G$ that fuse according to the group law of $G$. They correspond to the standard codimension-1 topological symmetry defects generating the symmetry $G$.

A general object may be expressed as a sum

$$\Omega = \bigoplus_{g \in G} \mathrm{Mod}(Z(\mathsf{C}_g)), \tag{154}$$

where the $\mathsf{C}_g$ form a collection of multi-fusion categories indexed by $g \in G$. This corresponds to stacking the elementary symmetry defects with 3d TQFTs of Turaev-Viro type. Choosing $\mathsf{C}_g = \mathrm{Vec}^{n_g}$ reproduces a direct sum of $n_g$ copies of symmetry defects labelled by $g$, similar to two and three dimensions. However, there are more general objects in four dimensions.

## 4.2 Gauging groups

Now consider gauging the symmetry $G$. The resulting symmetry 3-category is expected to be $3\mathrm{Rep}(G)$. There are a number of different interpretations of $3\mathrm{Rep}(G)$:

- It captures condensation defects for the topological Wilson surfaces in $\mathcal{T}/G$.

- It captures topological defects in $\mathcal{T}/G$ obtained by coupling to a 3-dimensional fully extended TQFT with symmetry $G$. This corresponds to a definition of $3\mathrm{Rep}(G)$ as the 3-category of 3-pseudo-functors

$$G \to 3\mathrm{Vec}, \tag{155}$$

  where $G$ is understood here as a strict 3-group, namely a 3-category with a single object, all of whose morphisms are invertible.

- It captures topological defects in $\mathcal{T}/G$ defined by topological defects in the original theory $\mathcal{T}$ together with instructions for how they intersect with networks of $G$ symmetry defects. This corresponds to a definition of $3\mathrm{Rep}(G)$ as bimodules for a certain 3-algebra object in $3\mathrm{Vec}(G)$. The construction must now take as input all possible topological defects in the original theory $\mathcal{T}$ of the form (154).

If one restricts attention to $\mathsf{C}_g = \mathrm{Vec}^{n_g}$, the classification of 3-representations is a straightforward generalisation of previous sections: an $n$-dimensional 3-representation is labelled by a permutation representation $\rho : G \to S_n$ and a 3-cocycle $c \in Z^3(G, U(1)^n)$, where $U(1)^n$ is understood as a $G$-module.

The simple 3-representations of this type then correspond to those for which $\rho$ is transitive and are induced by 1-dimensional 3-representations of subgroups of $G$ [89]. In this case, we can label simple 3-representations of $G$ by pairs consisting of

1. a subgroup $H \subset G$,

2. a class $c \in H^3(H, U(1))$.

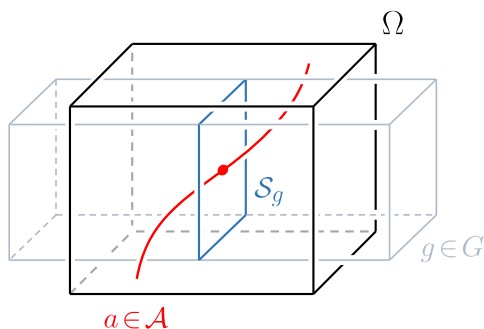

Figure 36

Physically, this corresponds to a codimension-1 defect on which the gauge symmetry is broken down to $H \subset G$ and supplemented by a topological action $c$.

However, there are more general 3-representations allowing for more general objects $3\mathrm{Vec}(G)$. Let us consider starting from an object $\Omega$ in $\mathcal{T}$ corresponding to a general combination of symmetry defects stacked with a 3d TQFT of Turaev-Viro type determined by a fusion category C. Equipping this with instructions for how to interact with networks of symmetry defects defines a class of one-dimensional irreducible 3-representations in $\mathcal{T}/G$, which reproduces the classification of $G$-graded extensions of fusion categories [90].

In more detail, we must provide a $G$-action on the braided fusion category $Z(\mathsf{C})$. There is then a sequence of obstructions to defining a consistent coupling of symmetry defects $g \in G$ to $\Omega$. Note that intersections with codimension-1 defects labeled by $g \in G$ take the form of 2-dimensional surfaces $\mathcal{S}_g$ on $\Omega$ as illustrated schematically in figure 36. Then the obstructions may be formulated as follows:

- The surface defects $\mathcal{S}_g$ may form a non-trivial 2-group with the simple abelian lines $\mathcal{A}$ in $Z(\mathsf{C})$. This is determined by an action $\rho : G \to \mathrm{Aut}(\mathcal{A})$ and a Postnikov class in $H^3(G, \mathcal{A})$. A non-trivial Postnikov class does not allow a consistent gauging of $G$, and this provides the first obstruction.

- If the first obstruction vanishes, there is a second obstruction from a possible 't Hooft anomaly for the surfaces $\mathcal{S}_g$ on $\Omega$, which is a class in $H^4(G, U(1))$.

If these obstructions vanish, one may consistently couple networks of symmetry defects to $\Omega$, which leads to irreducible 3-representations determined by a symmetry fractionalisation in $H^2(G, \mathcal{A})$ and an SPT phase $H^3(G, U(1))$.

More generally, irreducible 3-representations are labelled by a subgroup $H \subset G$ and a $H$-graded extension of a fusion category C, as discussed above. If we restrict attention to $\mathsf{C} = \mathrm{Vec}$, this reduces to the elementary 3-representations labelled by $H \subset G$ and $c \in H^3(H, U(1))$.

In the following, we will also have cause to consider projective 3-representations of $G$. They can arise at interfaces between theories $\mathcal{T}/G$ and $\mathcal{T}/_\phi G$, where $\phi \in H^4(G, U(1))$. In constructing such interfaces, one must couple to an obstructed $H$-action on a fusion category C, where the second obstruction should not vanish but match $\phi$. Such projective 3-representations can appear when gauging subgroups of $G$.

## 4.3 Gauging subgroups

Let us now consider a more general situation where the 0-form symmetry $G$ of $\mathcal{T}$ has an 't Hooft anomaly with representative $\alpha \in Z^5(G, U(1))$. Then the corresponding symmetry category $3\mathrm{Vec}^\alpha(G)$. This includes simple objects labelled by group elements $g \in G$ and fusion twisted by the 5-cocycle $\alpha$.

If $[\alpha]$ is trivial upon restriction to a subgroup $H \subset G$, the subgroup may be gauged. This requires choosing a trivialisation $\psi \in C^4(H, U(1))$ such that $\alpha|_H = (d\psi)^{-1}$, which is a generalisation of discrete torsion. We may then gauge $H$ by summing over networks of $H$-defects with phases $\psi(h_1, h_2, h_3, h_4)$ attached to junctions of four codimension-1 defects labelled by $H$.

As before, the topological defects in the gauged theory $\mathcal{T}/_\psi H$ are constructed from topological defects in the ungauged theory $\mathcal{T}$ together with instructions for how networks of $H$-defects may end on them consistently. This identifies topological defects after gauging with 3-bimodules for the algebra object $A(H, \psi)$ in $3\mathsf{Vec}^\alpha(G)$ associated to $H$ and $\psi$. We again denote the resulting symmetry category by $\mathsf{C}(G, \alpha \,|\, H, \psi)$.

Following through the arguments of the previous sections, the simple objects are labelled by pairs consisting of

1. a double coset $[g] \in H\backslash G/H$ with representative $g \in G$,

2. an irreducible projective 3-representation of $H_g := H \cap {}^g H \subset H$ with 4-cocycle

$$c_g(h_1, h_2, h_3, h_4) = \frac{\psi(h_1^g, h_2^g, h_3^g, h_4^g)}{\psi(h_1, h_2, h_3, h_4)} \cdot \frac{\alpha(h_1, h_2, h_3, h_4, g)\,\alpha(h_1, h_2, g, h_3^g, h_4^g)\,\alpha(g, h_1^g, h_2^g, h_3^g, h_4^g)}{\alpha(h_1, h_2, h_3, g, h_4^g)\,\alpha(h_1, g, h_2^g, h_3^g)} \,. \tag{156}$$

They depend on the choice of double coset representative $g$ and cocycle representative $c_g$ only up to isomorphism.

The irreducible projective 3-representations may be given a further explicit description recycling the discussion above: For those projective 3-representations that may be obtained by induction from a 1-dimensional one, specify a subgroup $K \subset H \cap {}^g H$ together with a 3-cochain $\phi \in C^3(K, U(1))$ satisfying $d\phi = c_g|_K$. However, similarly to above, we may also encounter projective 3-representations built by coupling to braided fusion categories with the appropriate projective G-action.

## 4.4 Gauging 3-subgroups

The most general situation we want to consider in four dimensions is a theory $\mathcal{T}$ with a finite 3-group symmetry $\mathcal{G}$. This is specified by a 0-form symmetry $K$, an abelian 1-form symmetry $A[1]$, and an abelian 2-form symmetry $C[2]$, together with actions of $K$ on both $A$ and $C$ and various Postnikov data. The latter may be summarised by cohomology classes[8]

$$\begin{aligned}
[e_3] &\in H^4(X, C), \\
[e_2] &\in H^3(K, A),
\end{aligned} \tag{157}$$

where $X$ denotes the 2-group formed by $A$ and $K$ with Postnikov class $[e_2] \in H^3(K, A)$.

The symmetry may have an 't Hooft anomaly specified by a class with representative $\mu \in Z^5(\mathcal{G}, U(1))$. The corresponding symmetry category is given by

$$3\mathsf{Vec}^\mu(\mathcal{G}). \tag{158}$$

The ambition is then to gauge an anomaly free 3-subgroup $\mathcal{H} \subset \mathcal{G}$ with a choice of trivialisation $\mu|_{\mathcal{H}} = (d\nu)^{-1}$ where $\nu \in C^4(\mathcal{G}, U(1))$. The outcome will be a 3-group-theoretical fusion 3-category

$$\mathsf{C}(\mathcal{G}, \mu \,|\, \mathcal{H}, \nu). \tag{159}$$

---

[8]In this section we introduce some additional indices in the Postnikov classes, in order to distinguish the two classes needed to specify the Postnikov data of a 3-group. We refer to appendix A for more details.

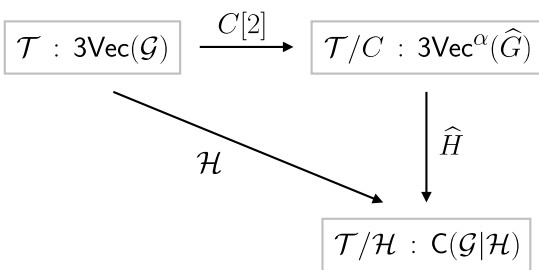

Figure 37

We will not attempt a general analysis here but leverage the above results on gauging subgroups together with some additional information about gauging 1-form symmetries to examine some special cases.

Let us suppose that the anomaly does not obstruct gauging the 2-form symmetry $C[2]$. This results in a theory $\mathcal{T}/C$ with a 2-group symmetry $\widehat{C} \rtimes X$ and mixed anomaly determined by the Postnikov data $[e_3] \in H^4(X,C)$. Then gauging general 3-subgroups may then be reduced to gauging 2-subgroups of $\widehat{C} \rtimes X$ analogously to section 3.4.

However, in general it is not possible to reduce the problem to gauging subgroups of ordinary groups, since gauging the 1-form symmetry will lead to another 1-form symmetry. The associated symmetry categories must therefore be studied independently. An exception is where the 1-form symmetry $A[1]$ is trivial, which is our first example below. Our second example is to independently gauge a 1-form symmetry. These results will then feed into the two case studies at the end of this section.

### 4.4.1   Example: no 1-form symmetry

Let us begin by considering the case where the 1-form symmetry of $\mathcal{G}$ is trivial, such that the Postnikov data reduces to a class

$$[e_3] \in H^4(K,C). \tag{160}$$

We are then interested in gauging an anomaly-free 3-subgroup $\mathcal{H} \subset \mathcal{G}$. This consists of subgroups $L \subset K$ and $D \subset C$ such that the group action of $K$ on $C$ restricts to a group action of $L$ on $D$ and $e_3|_L \in Z^4(L,C)$ is valued in $D$.

Let us assume the 't Hooft anomaly does not obstruct gauging the whole 2-form symmetry $C[2]$. In this case, $C[2]$ may be gauged first to obtain an ordinary group symmetry $\widehat{G} = \widehat{C} \rtimes K$ with mixed anomaly, to which we can then apply the machinery from previous subsections. Let us illustrate this procedure by gauging a 3-subgroup $\mathcal{H} \subset \mathcal{G}$ of an anomly-free 3-group $\mathcal{G}$ without discrete torsion. The two steps of the gauging procedure are then summarised in figure 37.

- First, we gauge $C[2]$ without discrete tosion to obtain a theory $\mathcal{T}/C$ with symmetry group $\widehat{G} = \widehat{C} \rtimes K$ and 't Hooft anomaly $\alpha$ represented by the SPT phase,

$$\int_X \widehat{c} \cup k^*(e_3), \tag{161}$$

in terms of the background fields $\widehat{c} \in H^1(X,\widehat{C})$ and $k : X \rightarrow BK$ for the $\widehat{G}$ symmetry. The symmetry category of $\mathcal{T}/C$ is therefore given by

$$\mathsf{C}(\mathcal{G}\,|\,C) = 3\mathsf{Vec}^\alpha(\widehat{G}). \tag{162}$$

- Next, we note that analogously to section 3.4 there is a 1-1 correspondence between 3-subgroups $\mathcal{H} \subset \mathcal{G}$ and subgroups $\widehat{H} \subset \widehat{G}$ with $\alpha|_{\widehat{H}} = 1$ given by

$$\mathcal{H} = (L, D) \quad \longleftrightarrow \quad \widehat{H} = \widehat{C/D} \rtimes L. \tag{163}$$

Gauging the 3-subgroup $\mathcal{H}$ in $\mathcal{T}$ can thus be achieved by gauging the subgroup $\widehat{H}$ in $\mathcal{T}/C$ using techniques described in subsection 4.3. The symmetry category of $\mathcal{T}/\mathcal{H}$ is therefore given by

$$\mathsf{C}(\mathcal{G}\,|\,\mathcal{H}) = \mathsf{C}(\widehat{G}, \alpha\,|\,\widehat{H}). \tag{164}$$

The situation is more involved when there is a non-trivial 1-form symmetry $A[1]$, since gauging $C[2]$ results in an anomalous 2-group. The dependence of the anomaly on the Postnikov data is expected to be a general feature. We study an example of this case study I below, which generalises slightly the examples proposed in [44].

### 4.4.2 Example: only 1-form symmetry

Let us now consider a theory $\mathcal{T}$ with an anomaly-free 1-form symmetry $A[1]$. The symmetry category is $3\mathsf{Vec}(A[1])$. This contains simple objects corresponding to condensation defects for the topological line operators generating $A[1]$, which correspond to fusion 2-categories

$$\mathsf{Mod}(\mathsf{Vec}(A)), \tag{165}$$

where $\mathsf{Vec}(A)$ is regarded as a braided fusion category with trivial braiding. However, there are again more general objects by combining with objects of the form $\mathsf{Mod}(Z(\mathsf{C}))$ for some fusion category $\mathsf{C}$.

Now consider gauging the symmetry $A[1]$. The symmetry category is expected to be $3\mathsf{Rep}(A[1])$. This contains objects corresponding to condensations for topological Wilson lines, which correspond to fusion 2-categories

$$2\mathsf{Rep}(A) = \mathsf{Mod}(\mathsf{Rep}(A)). \tag{166}$$

It is known that this reproduces a 1-form symmetry $\widehat{A}$, and therefore the symmetry category should also be equivalent to $3\mathsf{Vec}(\widehat{A}[1])$. This is compatible with the statements above because $\mathsf{Rep}(A) \cong \mathsf{Vec}(\widehat{A})$ as fusion categories.

We may also consider projective 3-representations of a 1-form symmetry $A[1]$. These would arise at three-dimensional interfaces between theories $\mathcal{T}/A[1]$ and $\mathcal{T}/_\phi A[1]$ for some $\phi \in H^4(A[1], U(1))$ and correspond to objects in $3\mathsf{Rep}^\phi(A[1])$. These associated topological defects are constructed by gauging $A[1]$ while coupling to a 3d TFQT with 1-form symmetry $A$ and 't Hooft anomaly $\phi \in H^4(A[1], U(1))$.

We caution that in the current setup such 3d TQFTs should be drawn from objects of the underlying category $3\mathsf{Vec}$, which are necessarily of Turaev-Viro type and therefore may not supply interesting projective 3-representations. A more interesting setup would therefore require an extension of $3\mathsf{Vec}$ to incorporate more general 3d TQFTs of the type considered in [91].

## 4.5 Case study I

Let us now consider a theory $\mathcal{T}$ with anomaly-free 3-group symmetry $\mathcal{G}$ with trivial 0-form symmetry component. This is specified by an abelian 1-form symmetry $A[1]$, an abelian 2-form symmetry $C[2]$, and Postnikov data

$$[e] \in H^4(B^2 A, C) \cong \mathrm{Hom}(\Gamma(A), C), \tag{167}$$

where $\Gamma(A)$ denotes the universal quadratic group of $A$. Gauging the entire 3-group symmetry results in a theory $\mathcal{T}/\mathcal{G}$ with symmetry category $3\mathrm{Rep}(\mathcal{G})$. A convenient method to uncover the structure of this symmetry category is gauging the 3-group in steps by first gauging $C[2]$ and then subsequently gauging $A[1]$:

- First gauging the 2-form symmetry $C[2]$ results in a theory $\mathcal{T}/C[2]$ with split 2-group symmetry $\widehat{\mathcal{G}} = \widehat{C} \times A[1]$ and mixed 't Hooft anomaly $\alpha$ represented by

$$\int_X \widehat{c} \cup e(\mathsf{a}) \tag{168}$$

  in terms of the background fields $\widehat{c} \in H^1(X, \widehat{C})$ and $\mathsf{a}: X \to B^2 A$ for the $\widehat{C} \times A[1]$ symmetry. We can denote the symmetry category $3\mathrm{Vec}^{\alpha}(\widehat{\mathcal{G}})$.

- Subsequently gauging $A[1]$ results in the symmetry category $3\mathrm{Rep}(\mathcal{G})$. Starting from $\mathcal{T}/C[2]$, the simple objects after gauging $A[1]$ are labelled by pairs:

  1. a character $\chi \in \widehat{C}$,
  2. a projective 3-representation of $A[1]$ with 4-cocycle $\langle \chi, e \rangle \in H^4(B^2 A, U(1))$.

  This captures the fact that the symmetry defects labelled by $\chi \in \widehat{C}$ in $\mathcal{T}/C[2]$ support an anomaly $\langle \chi, e \rangle \in H^4(B^2 A, U(1))$. This must be cancelled when gauging $A[1]$ by dressing with a three-dimensional TQFT with 1-form symmetry $A[1]$ whose anomaly cancels the one above.

The above is reminiscent of the dressing phenomenon appearing in [33] in terms of projective 3-representations and is a higher version of the appearance of projective representations of a quotient in the representation theory of group extensions, as summarised in section 2.4. Unfortunately, as mentioned above, $3\mathrm{Vec}$ is not rich enough to incorporate the full spectrum of topological defects constructed in [33].

## 4.6 Case study II

Let us consider a theory $\mathcal{T}$ in four dimensions with split 2-group symmetry $\mathcal{G} = D_4[1] \rtimes \mathbb{Z}_2$. We consider gauging 2-subgroups $\mathcal{H} \subset \mathcal{G}$. For simplicity, we omit a discussion of gauging with discrete torsion here.

An example is a pure $Spin(N)$ gauge theory with $N = 4k$, where $D_4 = \mathbb{Z}_2 \times \mathbb{Z}_2$ is the 1-form centre symmetry $Z(Spin(N))$ and $\mathbb{Z}_2$ is the outer automorphism group of $Spin(N)$ or charge conjugation. Gauging 2-subgroups will then allow us to determine the symmetry categories of global forms of four dimensional gauge theories with gauge algebra $\mathfrak{so}(N)$, including those with disconnected gauge groups.

We follow a similar notation for generators of the 2-group. We denote the generator of the 0-form symmetry by $s$ with $s^2 = 1$ and the generators of the 1-form symmetry by $a, b$ with $a^2 = b^2 = 1$.

- Consider gauging the subgroup $\mathcal{H} = \langle s \rangle \cong \mathbb{Z}_2$. This produces the $Pin^+(N)$ theory with symmetry category $3\mathrm{Rep}(D_4 \rtimes \mathbb{Z}_2)$.

- Consider the 2-subgroup $\langle ab \rangle[1] \subset G$ forming a non-split exact sequence of 2-groups

$$1 \to \mathbb{Z}_2[1] \to \mathcal{G} \to \mathbb{Z}_2[1] \times \mathbb{Z}_2 \to 1, \tag{169}$$

with extension class in $H^3(B^2\mathbb{Z}_2 \times B\mathbb{Z}_2, \mathbb{Z}_2)$ represented by

$$\frac{1}{2} \int \mathsf{a}' \cup \mathsf{b}. \tag{170}$$

Here we introduce background fields satisfying $\delta\mathsf{a} = \mathsf{a}' \cup \mathsf{b}$. Gauging this 2-subgroup therefore generates a 2-group symmetry $\mathbb{Z}_2[1] \times (\mathbb{Z}_2[1] \times \mathbb{Z}_2) \cong D_4[1] \times \mathbb{Z}_2$ with cubic 't Hooft anomaly $\alpha$ represented by the SPT phase

$$\frac{1}{2} \int_X \widehat{\mathsf{a}} \cup \mathsf{a}' \cup \mathsf{b}. \tag{171}$$

This is the $SO(N)$ gauge theory with symmetry category $2\mathsf{Vec}^\alpha(D_4[1] \times \mathbb{Z}_2)$.

- Consider gauging the subgroup $D_4[1] = \langle a, b \rangle[1]$ This results in the $PSO(N)$ theory with anomaly free 2-group symmetry $D_4[1] \rtimes \mathbb{Z}_2$.

- Consider gauging the 2-subgroup $\mathbb{Z}_2[1] \times \mathbb{Z}_2 = \langle ab \rangle[1] \times \langle s \rangle$. This reproduces the $O(N)$ gauge theory. The 't Hooft anomaly of $SO(N)$ obtained after gauging $\langle ab \rangle[1]$ now translates into a 3-group symmetry

$$\mathbb{Z}_2[2] \times_e D_4[1], \tag{172}$$

with 2-form symmetry $\mathbb{Z}_2[2]$, 1-form symmetry $D_4[1]$ and a non-trivial Postnikov class $[e] \in H^4(B^2 D_4, \mathbb{Z}_2)$ such that the background fields satisfy

$$\delta\widehat{\mathsf{b}} = \widehat{\mathsf{a}} \cup \mathsf{a}'. \tag{173}$$

The symmetry category is therefore $3\mathsf{Vec}(\mathbb{Z}_2[2] \times_e D_4[1])$.

- Consider gauging $\langle a \rangle$ or $\langle b \rangle$. These correspond to $Ss(N)$ and $Sc(N)$ gauge theories respectively. They can be obtained from $SO(N)$ by gauging the $(\mathbb{Z}_2 \times \mathbb{Z}_2)[1]$ symmetry, or equivalently by starting from the $O(N)$ theory above and gauging the entire 3-group. From this perspective, the simple objects are labelled by elements $\chi \in \mathbb{Z}_2$ and projective 3-representations of $(\mathbb{Z}_2 \times \mathbb{Z}_2)[1]$ with 4-cocyle $\langle \chi, e \rangle$, where $e$ is the element of $H^4(B^2(\mathbb{Z}_2 \times \mathbb{Z}_2), \mathbb{Z}_2)$ represented by

$$\frac{1}{2} \int \widehat{\mathsf{a}} \cup \mathsf{a}'. \tag{174}$$

The symmetry category is $3\mathsf{Rep}(\mathbb{Z}_2[2] \times_e D_4[1])$.

- Gauging the whole 2-group gives the $PO(N)$ gauge theory whose symmetry category is therefore $3\mathsf{Rep}(D_4 \rtimes \mathbb{Z}_2)$, equivalent to that of $Pin^+(N)$.

These results are summarised in figure 38.

Finally, we note that a number of these symmetry categories are transformed to an equivalent symmetry category under gauging a 1-form symmetry, in a manner that is compatible with S-duality. Indeed, by an argument to the $c = 1$ CFT discussed in section 2.6, this leads to additional non-invertible duality defects at specific values of the coupling where theories are invariant under gauging [32, 33, 43, 44, 64].

# Acknowledgments

The authors would like to thank Alex Bullivant, Lea Bottini, Thibault Décoppet, Iñaki García-Etxebarria, Saghar Hosseini, and Sakura Schäfer-Nameki for discussions. We especially thank Lakshya Bhardwaj for drawing to our attention some inaccuracies in section 4 of the first version.

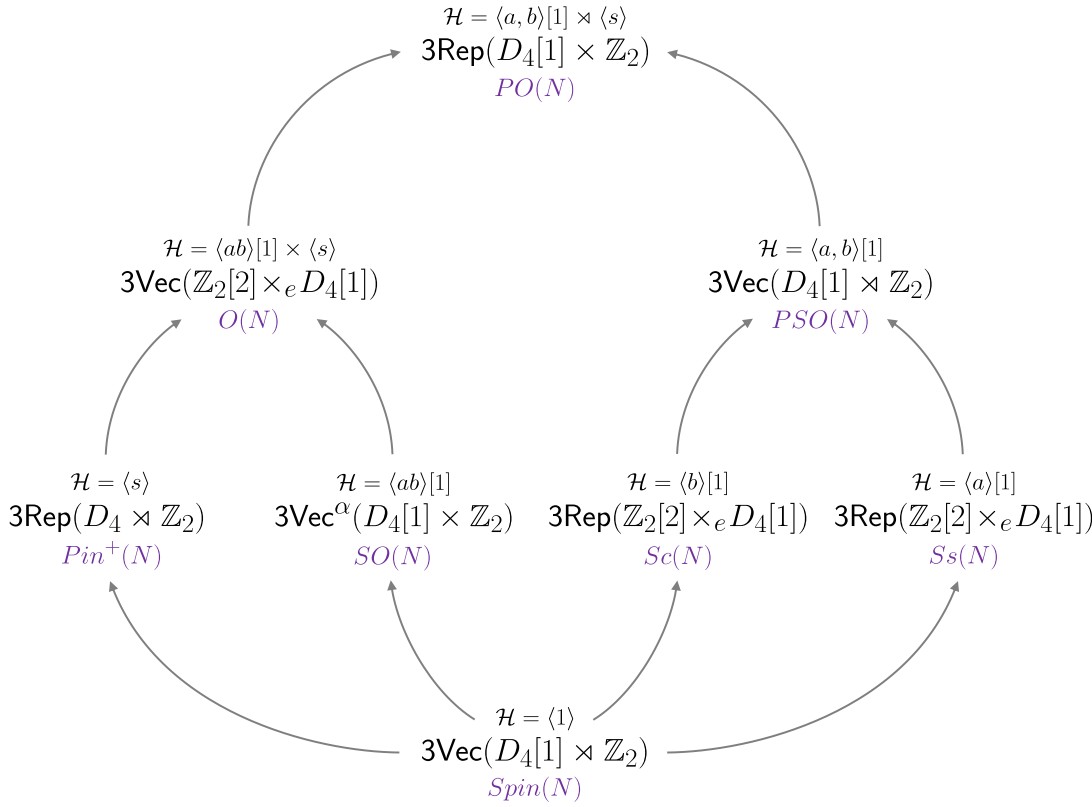

Figure 38

**Funding information** The work of MB and AF is supported by the EPSRC Early Career Fellowship EP/T004746/1 "Supersymmetric Gauge Theory and Enumerative Geometry". MB is also supported by the STFC Research Grant ST/T000708/1 "Particles, Fields and Spacetime", and the Simons Collaboration on Global Categorical Symmetry.

# A  Spectral sequences

We first start with a theory in $D$ dimensions with an ordinary 0-form symmetry $G$ given by the central extension

$$A \to G \to K \simeq G/A. \tag{A.1}$$

The classifying data for this extension is a Postnikov class $e \in H^2(K, A)$, and when this vanishes we have simply $G = A \times K$. One way to gauge $G$ is via a sequence of gaugings

$$(D-1)\mathsf{Vec}(G) \xrightarrow{A} (D-1)\mathsf{Rep}(A) \times (D-1)\mathsf{Vec}(K) \xrightarrow{K} (D-1)\mathsf{Rep}(G). \tag{A.2}$$

The symmetry category $(D-1)\mathsf{Rep}(A)$ in the intermediate theory will ultimately include the symmetry $\widehat{A}[D-2]$. This $(D-2)$-form symmetry has a mixed anomaly with $K$ corresponding to the $(D+1)$-dimensional SPT phase

$$\int \widehat{\mathsf{a}} \cup \mathsf{k}^*(e). \tag{A.3}$$

Next we consider those SPT phases we could include while gauging $G$ classified by $H^D(G, U(1))$. The exact sequence A.1 determines a Lyndon-Hochschild-Serre spectral se-

quence that approximates SPT phases

$$E_2^{p,q} = H^p(K, H^q(A, U(1))) \Rightarrow H^{p+q}(G, U(1)). \tag{A.4}$$

Certainly this spectral sequence and all that follows can still be described when $A \triangleleft G$ is a generic abelian normal subgroup of $G$, we just need to include that the group cohomology is twisted by an action of $K$ on $A$. For the sake of simplicity however we will restrict ourselves to central extensions.

## A.1 Split central extensions

An important fact for calculation is that when $G = A \times K$, the spectral sequence collapses at the $E_2$-page and reduces to a decomposition

$$H^D(G, U(1)) = \bigoplus_{p+q=D} H^p(K, H^q(A, U(1))). \tag{A.5}$$

We note that each term appearing in the decomposition above corresponds to a choice we can make in the gauging sequence A.2.

The most obvious two examples are the pure SPT phases for $A$ and $K$, which correspond to the pages $E_2^{0,D} = H^D(A, U(1))$ and $E_2^{D,0} = H^D(K, U(1))$ respectively.

Another important example is the page $E_2^{D-1,1} = H^{D-1}(K, \widehat{A})$ which corresponds to a choice of symmetry fractionalisation of $K$ by $\widehat{A}[D-2]$ in the intermediate theory of A.2.

The other terms in the decomposition correspond to other symmetry fractionalisations of $K$ by condensation defects appearing in $(D-1)\mathrm{Rep}(A)$.

## A.2 Obstructions

When the Postnikov class $e$ is non-trivial we instead find that there are obstructions to lifting the decomposition A.5 to a class in $H^d(G, U(1))$. Finding terms for which there are no obstructions takes us to higher pages in the spectral sequence defined cohomologically as

$$\begin{aligned} d_r^{p,q} &: E_r^{p,q} \to E_r^{p+r,q-r+1}, \\ d_r^2 &= 0, \\ E_{r+1} &= H(E_r, d_r). \end{aligned} \tag{A.6}$$

The differential $d_r$ generates these obstructions and depends on the Postnikov class $e$. We notice now that these differentials map pages that approximate SPT phases in $d$ dimensions to pages that approximate SPT phases in $(d+1)$ dimensions. In other words, these obstructions describe "trivial" $d$-dimensional 't Hooft anomalies for $G$ that we can cancel with an SPT phase.

We also note that the spectral sequence obstructions correspond to anomalies and extensions in A.2 that would obstruct gauging the full sequence.

Obvious examples include the final obstructions generated by $d_r^{D+1-r,r-1}$ which are all valued in $H^{D+1}(K, U(1))$ and correspond to pure $K$ 't Hooft anomalies that obstruct the second step of the gauging sequence.

A more interesting class of obstructions is the one before the final obstruction generated by $d_{r-1}^{D+1-r,r-1}$ which are valued in $H^D(K, \widehat{A})$ which corresponds to a non-trivial Postnikov class for a $(D-1)$-group with 0-form part $K$ and $(D-2)$-form part $\widehat{A}[D-2]$. This would obstruct the gauging sequence by making it impossible to gauge $K$ independently of $\widehat{A}[D-2]$. We note that these obstructions also correspond to symmetry fractionalisations in $(D+1)$ dimensions; if SPT phases in one dimension higher correspond to 't Hooft anomalies on the boundary, then symmetry fractionalisations in one dimension higher correspond to non-trivial extensions on the boundary.

The other obstructions that can appear correspond to other Postnikov classes that describe higher groups with $K$ as the 0-form part and condensation defects from $(D-1)\text{Rep}(A)$ appearing as higher-form parts. These types of obstructions together with the classes of obstructions above taken together describe a general extension of symmetry categories that we might write as

$$(D-1)\text{Rep}(A) \rightarrow \mathcal{C} \rightarrow (D-1)\text{Vec}(K). \tag{A.7}$$

Such extensions of fusion $(D-1)$-categories and their classification are not well documented in the maths literature and so this represents a new and exciting direction of research for gauging (higher) subgroups.

## A.3 Postnikov systems and general spectral sequences

We can extend this formalism to more general group-like symmetries $D$ dimensions relatively easily. Suppose we have a $(D-1)$-group $\mathcal{G}$ whose components are finite and labelled by homotopy groups $\pi_n(B\mathcal{G})$ of an associated classifying space $B\mathcal{G}$ for $1 \leq n \leq D-1$. One way to construct such a classifying space comes courtesy of a Postnikov system, which is a sequence of fibrations

$$B^n \pi_n(B\mathcal{G}) \rightarrow X_n \rightarrow X_{n-1}, \quad 2 \leq n \leq D-1, \tag{A.8}$$

such that $X_{D-1} \simeq B\mathcal{G}$ and $X_1 \simeq B\pi_1(B\mathcal{G})$. These fibrations are classified by homotopy classes of maps

$$[e_n] \in [X_{n-1}, B^{n+1}\pi_n(B\mathcal{G})] \simeq H^{n+1}(X_{n-1}, \pi_n(B\mathcal{G})), \tag{A.9}$$

called Postnikov classes. Each fibration has an associated Leray-Serre spectral sequence that must each be computed in order to construct the de Rham cohomology $H^\bullet(B\mathcal{G})$. For example focus on a single fibration for $B^n \pi_n(\mathcal{G}) = B^n A$ over some $X_{n-1}$

$$B^n A \rightarrow X_n \rightarrow X_{n-1}. \tag{A.10}$$

To compute $H^\bullet(X_n)$ we have a spectral sequence with $E_2$-page

$$E_2^{p,q} \simeq H^p(X_{n-1}, H^q(B^n A)), \tag{A.11}$$

and to construct these pages we also need the fibration for $B^{n-1}\pi_{n-1}(B\mathcal{G})$ over $X_{n-2}$ which in turn comes with its own spectral sequence. This series of spectral sequence calculations then continues for each subsequent fibration in the Postnikov tower.

We might be concerned that this computation quickly becomes very complicated and ideally we would like an algebraic analogue for higher group cohomology, but at least we can restrict to classes of higher groups for which this calculation is more manageable and yet sufficiently rich to demonstrate the range of behaviour SPT phases for higher groups can describe. Just as was the case for ordinary subgroups, these spectral sequences should collapse at their respective $E_2$-page if the associated fibration splits.

## A.4 Spectral sequences for 't Hooft anomalies

We may think of a theories 't Hooft anomaly as an SPT phase in $(D+1)$-dimensions flowing to that theory placed on a $d$-dimensional boundary. The mixed anomaly of $K \times \widehat{A}[D-2]$ in the previous section is one such example, and the classifying space of this $(D-1)$-group is described by a (split) fibration

$$B^{D-1}\widehat{A} \rightarrow B(K \times \widehat{A}[D-2]) \rightarrow BK. \tag{A.12}$$

The associated spectral sequence for $H^{D+1}(B(K \times \widehat{A}[D-2]), U(1))$ collapses at the $E_2$-page. The mixed anomaly corresponds to an element in $E_2^{2,D-1} \simeq H^2(BK, A)$, which is exactly the

group that classifies extensions of $K$ by $A$. We might also say that the mixed anomaly is just the image of the Postnikov class under the spectral sequence.

We can also apply this logic to other Postnikov classes that might appear in a higher group symmetry. Take again our $(D-1)$-group symmetry example with $\pi_{D-1}(B\mathcal{G}) = A$, then provided the $(D-2)$-form is not anomalous we can gauge it. The Postnikov class $[e_{D-1}] \in H^D(X,A)$ appears in the $E_2^{D,1}$ page of the spectral sequence for $H^{D+1}(X_{D-2} \times B\widehat{A}, U(1))$. Its image is a mixed anomaly corresponding to the SPT phase

$$\int \widehat{\mathsf{a}} \cup \mathsf{x}^*(e_{D-1}), \tag{A.13}$$

where $\mathsf{x}$ is a collection of background fields for the remaining $(D-2)$-group classified by $X_{D-2}$.

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
