# Peer review of "Non-invertible Symmetries and Higher Representation Theory II"

_SciPost Physics, doi:SciPost Phys. 17, 067 (2024)_

## Round 1 · Referee Report · Anonymous · 2024-2-4

Report

The authors have addressed all my comments in a satisfactory way, thus I highly recommend publication.

---

## Round 1 · Referee Report · Anonymous · 2024-2-19

Strengths

1) The paper offers a clear mathematical perspective on several previously known physical constructions of non-invertible symmetries in D>2.

2) It clarifies what is the categorical characterization of several known group-theoretical non-invertible symmetries.

3) It discusses in detail many examples, so it is useful for the reader who wishes to reproduces the results.

Weaknesses

1) The paper is very long, extremely detailed, and many things are repeated several times without making a clear point.

2) The authors do not attempt to convince the reader that the explanation of what are the mathematical structures underling the known non-invertible symmetries has some relevance for physics. Hence the most physics oriented reader may have the feeling that the analysis is a purely academic exercise.

3) While the paper presents a general mathematical structure that seems to explain in a unified languale a varieties of phenomena known from a physics perspective, like the appearance of TQFT coefficients, and dressing of the defects with TQFT, the analysis shows that there is a mismatch (see report) between the mathematical formalism and the expectation from physics which the authors didn't attempt to fill or at least comment on.

Report

The paper analyzes systematically various group-theoretical higher-categories in D=2,3,4, giving a unified mathematical perspective on several previously known facts. It starts reviewing the D=2 case, where many things are well known, with a perspective which explains interesting phenomena discovered in the physics literature in higher D.

Several known non-invertible symmetries in D=3,4 are characterized as higher categories of higher-representations, and the authors discuss several examples which can help the reader to acquire familiarity with the type of computations involved.

One of the main interesting finding is the fact that, the dressing of topological defects with TQFT while gauging some (higher)subgroup which is known to be necessary in D>2, is already present in D=2: the anomaly of the bulk implies that the topological lines support a quantum mechanics with an anomaly, namely a projective representation, and one must stack a topological QM with an opposite anomaly while gauging. The necessity of stacking TQFT in D>2 is then understood similarly as higher-projective representations on the defect which must be cancelled with anomalous TQFTs.

However, while this clarifies previous findings in D=3, the situation in D=4 shows a clear mismatch between the mathematical formalism developed and previous finings in the physics literature. In particular the formalism developed in this paper seems to predict that the 3d TQFTs necessary for dressing topological defects while gauging are always of the Turaev-Viro type, but in many papers, starting with 2111.01141, it has been shown that there are simple examples (which should definitely be not beyond the formalism developed in the present paper) where one needs chiral TQFTs. While the authors makes some quick comment about this (e.g. at the end of section 4.5), I think this is a quite serious issue which would require ideally some extra analysis, or at least some more extended comment, explaining why and how the formalism developed here does not cover these cases.

Overall the paper is interesting and well written, and if the authors can fulfill the above main issue together with some other minor improvement I would advice the paper for publication.

I'll report here a list of comments, suggestions and questions, which the authors may want to consider in order to improve the presentation. In the "requested changes" I'll report instead things which I believe should be address before publication.

1) A comment on the second bullet point at pag. 6, which also applies to analogue discussions in D=3,4. The functors described are "fiber functors" and their existence is the categorical statement of the fact that the symmetry can be realized in a trivially gapped phase, i.e. it is anomaly free. Maybe it is worth explaining the relation with the first bullet point, to give a "physical" flavor of why anomaly free is equivalent to the existence of fiber functors.

2) The presentation in sec. 2.3.1 appears involved. Why do authors start with a definition of \Phi_g in eq. 2.11 and then successively switch to an alternative definition to get 2.14, instead of directly starting with the alternative? Same comment in D=3.

3) Most of the material in sec 2.4 is well known (in particular sec 2.4.1) and I would suggest to reduce it to improve readability.

4) Eq. (3.1) is confusing. I guess the authors have in mind placing the defects on S^2, otherwise the result of stacking depends on topology. I think that's the reason why after 3.1 they conclude the coefficients are integers instead of TQFTs. In section 4 they also comment that the appearance of TQFTs is peculiar of 3d defects in D=4 theories, but this is in contrast e.g. with the results of 2204.02407, I think precisely because the authors do not consider 2-manifolds other than spheres.

5) Beginning of sec. 3.4: H^3(K,A) should be twisted by \phi. Is this considered here?

6) Beginning of sec. 3.4: I'm not aware of a group-cohomology formulation of the cohomology of classifying spaces of 2-groups. For instance already for K trivial, the anomaly is classified by H^4(B^2A,U(1)). This has an algebraic formulation as the group of quadratic forms over A, but it is not of the group-cohomology type. The situation is even worst in higher D, and when the higher-group structure is non trivial, and as far as I know the anomaly is classified in general by something calculated by a spectral sequence. If the authors are aware of a simpler group-cohomology-like formulation it is worth reviewing it here.

7) Eq. 3.45 and below: the group cohomology should be twisted. It is not clear if the authors are using this or not.

Requested changes

1) Address somehow (either resolving it, or commenting about the mismatch) the above mention issue that the formalism contains a mismatch with previous physics analysis in D=4.

2) The notation in eq. (1.2) is not explained there, but only much below. I'd suggest to explain the notation every time a symbol appears for the first time.

3) Why is there a (-1) in eq. (2.3)? And in many other similar places. It is fine but doesn't match with the intro...

4) I guess there is a typo in the the first sentence of the second paragraph of section 2.6

5) An other mismatch (already present in the part I paper of the same authors) between the present mathematical formalism and the expectation from physics: physics intuition would suggest that condensates are labelled by subgroups instead of equivalence classes of them. It seems however that the math theory on 2reps produces the second option.

6) Typo first line second paragraph sec. 3.4: \hat{\mathcal{G}} ---> \hat{G}

7) Typo first line sec. 4.5: "now" repeated

---

## Round 1 · Author Response

We thank the referee for their detailed report and for pointing out instances with a need for clarification in our paper. Below, we address some of the questions raised in the report:

• Four dimensions: As the mathematical literature on fusion 3-categories is less developed, our discussion in four dimensions is intentionally less conclusive and systematic as in the previous sections. In particular, we do not wish to include Reshetikin-Turaev type theories in our discussion of 3d TQFTs, as was clarified between versions 1 and 2 of the paper.

• Simple objects in 3Vec: We are not aware of a sensible equivalence relation that could be imposed on objects in the 3-category 3Vec to render the number of simple objects finite. Similarly for 3Rep(G), for which 3Vec acts as a “coefficient system”.

---

## Round 1 · List of Changes

Below, we summarise the relevant changes that we made in order to address the main points raised in the report:

• Summary of results: Added eq. (1.2) to the introduction to summarise and highlight the main findings of the paper (the classification of symmetry defects after gauging).

• Euler terms: Clarified our treatment of Euler terms in a footnote on page 23. These fix the partition function of a decoupled 2d TQFT on a 2-sphere. If the Euler terms are non-trivial, the associated topological surface defect acts universally by a non-trivial scalar multiplication on all local operators, including the identity operator. Since topological defects of this type are not particularly interesting from a symmetry perspective, we only consider canonically normalised (stable) 2d TQFTs with trivial Euler terms in what follows.

• Projective 2-representations: Irreducible projective 2-representations of a group G with 3-cocycle u can be labelled by a choice of subgroup H ⊂ G together with a trivialisation dc = u|_H. For generic H, such a trivialisation may not exist, in which case there exists no corresponding projective 2-irrep labelled by H. We clarified this point in a footnote on page 25.

• Typos: fixed the following typos:
- Psi ∈ C^{D-1} --> Psi ∈ C^D on page 3 above eq. (1.1)
- Fixed undefined references on page 24

• Notation: Gave further motivation for the notation H_g := H ∩ gHg^{-1} for the subgroup of H that leaves a given graded line defect V_g invariant when intersecting it, see page 9.

---

## Round 2 · Referee Report · Anonymous (Referee 4) · 2024-5-14

Report

The authors have addressed most of questions, except for the point about TQFT dressing in D=4: they only find Turaev-Viro type TQFT, while it is well known that other TQFTs can appear. I will recommend the paper for publication provided the author will resolve or comment about this issue.

Recommendation

Ask for minor revision

---

## Round 2 · Author Response

We thank the referee for their detailed report and for pointing out instances with a need for clarification in our paper. Below, we address some of the questions raised in the report:

• Fibre functors: The second bullet point on page six is intended to give an alternative interpretation of the category Rep(G) for a finite group G, which may be interpreted as the category of functors F: BG -> Vec from the delooping of G into Vec. We refrain from calling these “fibre functors” as the latter term is usually reserved for cases where the pre-image of the functor is a (higher) fusion category corresponding to the symmetry category of a given theory (and not the delooping of a finite group, which is not a fusion category).

• Condensation defects: Condensation defects are (partly) labelled by subgroups: While simple objects in 2Rep(G) are labelled by pairs (H,c) consisting of a subgroup H ⊂ G and a class c ∈ H^2(H,U(1)) , two such simple objects (H,c) and (H’,c’) are considered equivalent if there exists a group element g ∈ G such that H’ = gHg^(-1) and c’ = c^g. From a physical point of view, a symmetry defect labelled by (H,c) corresponds to a surface where the bulk gauge symmetry G is broken down to H and supplemented by an SPT phase c. In particular, there is no physical distinction between surfaces obtained by gauging conjugate subgroups H and H’, since they only differ by a residual symmetry transformation g on the defect. This will be clarified in an updated version of Part I and is stated in the first paragraph of page 25 of Part II.

• Bimodule classification: The definition of \Phi_g as in eq. (2.11) is the natural one given the left- and right-morphisms l and r of a bimodule, and is also the standard definition appearing in the mathematical literature on the classification of such bimodules [24]. We implement the redefinition (2.13) to achieve a more canonical form of the 2-cocycle (2.14), but retain its original form for comparison.

• Cohomology of 2-groups: We are not aware of a group-cohomology classification of the cohomology of the classifying space of a 2-group. Our notation H^n(\mathcal{G}, U(1)) is explained in footnote 5.

• Gauging extensions in 2d: We included section 2.4 with the presented level of detail to make completely apparent the analogy with the gauging of sub-2-groups discussed in section 3.4 (in particular the analogy between figure 13 and figures 30/31).

---

## Round 2 · List of Changes

Below, we summarise the relevant changes that we made in order to address the main points raised in the report:

• TQFT coefficients: Given a surface defect X in three dimensions, one may in principle obtain new surface defects by stacking X with decoupled 2d TQFTs. However, since 2d fully extended stable TQFTs are (up to equivalence) completely classified by positive integers n (corresponding to their number of vacua), stacking such a TQFT T_n on top of X simply corresponds to taking the direct sum T_n ⊗ X = X ⊕ … ⊕ X = n ⋅ X. This is the nature of integer fusion coefficients in equations such as (3.1). In particular, the mathematical fusion rules are not associated to particular topologies of the surface defects but internal to the fusion 2-category under consideration. We clarified our treatment of Euler terms in a footnote on page 23.

• Typos:
- Added (−1) to \alpha|_H = (d\psi)^(−1) above eq. (1.1) in the introduction. We choose this convention to interpret \psi(h_1, h_2) as the phase assigned to the junction of two H-lines, as illustrated in figure 2.
- Further explained the notation used in eq. (1.2).
- Fixed typo (”now” repeated) in first sentence of section 4.5.
- Clarified our usage of twisted group cohomology by a footnote on page 35.
- Added “orbifold branch” to the first sentence of the second paragraph of section 2.6.

---

## Round 3 · Author Response

In this paper, we are only considering 3-dimensional TQFTs of Turaev-Viro type when constructing topological symmetry defects in four dimensions, where the mathematical literature is less well-developed. Incorporating more general TQFTs would require the development of substantial new theoretical background, which is beyond the scope of this paper and is left to future work. As a result, our formalism only captures a subset of the topological defects constructed for example in 2111.01141. We clarified this point in the last paragraph of subsection 1.1 and the third paragraph of section 4.

---

## Round 3 · List of Changes

• Commented on the fact that we are only considering dressing with 3d TQFTs of Turaev-Viro type in D=4 in the last paragraph of subsection 1.1 and the third paragraph of section 4 , as incorporating more general TQFTs is beyond the scope of this paper.

---

## Editorial Decision

published